

# Bias assessment of lower and middle tropospheric CO₂ concentrations of GOSAT/TANSO-FTS TIR Version 1 product

Naoko Saitoh[1], Shuhei Kimoto[1*], Ryo Sugimura[1**], Ryoichi Imasu[2], Kei Shiomi[3], Akihiko Kuze[3], Yosuke Niwa[4], Toshinobu Machida[5], Yousuke Sawa[4], and Hidekazu Matsueda[4]

[1]Center for Environmental Remote Sensing, Chiba University, Chiba, 263-8522, Japan
[2]Atmosphere and Ocean Research Institute, The University of Tokyo, Kashiwa, 277-8564, Japan
[3]Japan Aerospace Exploration Agency, Tsukuba, 305-8505, Japan
[4]Meteologotical Research Institute, Tsukuba, 305-0052, Japan
[5]National Institute for Environmental Studies, Tsukuba, 305-8506, Japan
[*]Now at the Japan Research Institute, Tokyo, 141-0022, Japan
[**]Now at Fujitsu FIP Corporation, Tokyo, 105-8668, Japan

*Correspondence to*: Naoko Saitoh (nsaitoh@faculty.chiba-u.jp)

**Abstract.** CO₂ observations in the free troposphere can be useful for constraining CO₂ source and sink estimates at the surface due to their representativeness being away from local point sources of CO₂. The thermal infrared (TIR) band of the Thermal and Near Infrared Sensor for Carbon Observation (TANSO)−Fourier Transform Spectrometer (FTS) on board the Greenhouse Gases Observing Satellite (GOSAT) has been observing global CO₂ concentrations in the free troposphere for about 8 years, and thus could provide a dataset with which to evaluate the vertical transport of CO₂ from the surface to the upper atmosphere. This study evaluated biases in the TIR version 1 (V1) CO₂ product in the lower troposphere (LT) and the middle troposphere (MT) (736−287 hPa), on the basis of comparisons with CO₂ profiles obtained over airports using Continuous CO₂ Measuring Equipment (CME) in the Comprehensive Observation Network for Trace gases by AIrLiner (CONTRAIL) project. Bias-correction values are presented for TIR CO₂ data for each pressure layer in the LT and MT regions during each season and in each latitude band: 40°S–20°S, 20°S–20°N, 20°N–40°N, and 40°N–60°N. TIR V1 CO₂ data had consistent negative biases of 1−1.5% compared with CME CO₂ data in the LT and MT regions, with the largest negative biases at 541−398 hPa, partly due to the use of 10-μm CO₂ absorption band in conjunction with 15-μm and 9-μm absorption bands in the V1 retrieval algorithm. Global comparisons between TIR CO₂ data to which the bias-correction values were applied and CO₂ data simulated by Nonhydrostatic ICosahedral Atmospheric Model (NICAM)-based transport model (TM) confirmed the validity of the bias-correction values evaluated over airports in limited areas. In low latitudes in the upper MT region (398−287 hPa), however, TIR CO₂ data in northern summer were overcorrected by these bias-correction values; this is because the bias-correction values were determined using comparisons mainly over airports in East Asia where CO₂ concentrations in the upper atmosphere display relatively large variations due to strong updrafts.



## 1. Introduction

$CO_2$ in the atmosphere is the most influential greenhouse gas (IPCC, 2013 and references therein). Many studies have been conducted to estimate the sources and sinks of atmospheric $CO_2$ using both observational data and transport models (e.g., Gurney et al., 2002; 2004). In $CO_2$ inversion studies, accurate atmospheric $CO_2$ observations with spatial representativeness are desirable, which can be obtained from elevated sites such as tall towers and mountains or over the ocean. Patra et al. (2006) demonstrated the robustness of $CO_2$ surface flux estimation using $CO_2$ data obtained solely from ocean sites compared to data obtained from both ocean and land sites; this was because the models discussed therein were unable to successfully simulate $CO_2$ data over land, as these sites were more affected by local point sources of $CO_2$.

Uncertainties in atmospheric transport processes also result in differences in $CO_2$ surface fluxes estimated by inverse models. $CO_2$ is chemically inactive, and thus long-range transport processes as well as surface fluxes determine its horizontal distribution and seasonal cycle in the atmosphere (Miyazaki et al., 2008; Barnes et al., 2016). The treatment of vertical transport of $CO_2$ also produces differences in simulated $CO_2$ concentrations in the free troposphere among transport models unrelated to surface fluxes (Niwa et al., 2011a). Therefore, it is needed to observe $CO_2$ concentrations over land that are not strongly affected by local point sources of $CO_2$ emissions, as well as $CO_2$ concentrations in the free troposphere that can evaluate vertical $CO_2$ transport from the surface in transport models.

Satellite-borne nadir-viewing sensors can observe averaged $CO_2$ concentrations, with horizontal resolution ranging from several kilometers to tens of kilometers. Column-averaged dry-air mole fractions of $CO_2$ ($XCO_2$) have been observed utilizing $CO_2$ absorption bands in the shortwave infrared (SWIR) regions at around 1.6 and/or 2.0 μm by satellite-borne sensors such as the Scanning Imaging Absorption Spectrometer for Atmospheric Chartography (SCIAMACHY) on the Environmental Satellite (ENVISAT) (Buchwitz et al., 2005; Barkley et al., 2006), the Thermal and Near Infrared Sensor for Carbon Observation (TANSO)–Fourier Transform Spectrometer (FTS) on the Greenhouse Gases Observing Satellite (GOSAT) (Yoshida et al., 2011, 2013; O'Dell et al., 2012; Butz et al., 2011; Cogan et al., 2012), and the Orbiting Carbon Observatory 2 (OCO-2) (Crisp et al., 2017; Connor et al., 2016). Global $XCO_2$ data based on satellite observations are averaged concentrations in their field of views of several kilometers that are not too much affected by strong local point sources of $CO_2$, and have therefore been used to estimate surface $CO_2$ fluxes (Maksyutov et al., 2013; Saeki et al., 2013a; Chevallier et al., 2014; Basu et al., 2013, 2014; Takagi et al., 2014). $CO_2$ concentrations in the free troposphere can be obtained by satellite-borne sensors with thermal infrared (TIR) bands at around 4.6, 10, and/or 15 μm, provided by the following sensors: the High-Resolution Infrared Sounder (HIRS) (Chédin et al., 2002, 2003, 2005), the Interferometric Monitor for Greenhouse Gases (IMG) (Ota and Imasu, 2016), the Atmospheric Infrared Sounder (AIRS) (Crevoisier et al., 2004; Chahine et al., 2005; Maddy et al., 2008; Strow and Hannon, 2008), the Tropospheric Emission Spectrometer (TES) (Kulawik et al., 2010, 2013), the Infrared Atmospheric Sounding Interferometer (IASI) (Crevoisier et al., 2009), and the TANSO-FTS (Saitoh et al., 2009, 2016). Furthermore, $CO_2$ concentrations in several atmospheric layers within the free troposphere can be retrieved separately from high-resolution TIR spectra (Saitoh et al., 2009; Kulawik et al., 2013). Such





vertical $CO_2$ data offer a good constraint for $CO_2$ surface flux estimates (Kulawik et al., 2010), and have the potential to evaluate the vertical transport of $CO_2$ from the surface to the upper atmosphere, if they have sufficient accuracy.

Previously, the data quality of $CO_2$ product from the GOSAT/TASNO-FTS TIR band has been examined in the upper troposphere and the lower stratosphere (UTLS) region, where TIR observations have the most sensitivity to $CO_2$

concentrations. Saitoh et al. (2016) evaluated biases in UTLS (287−162 hPa) $CO_2$ data of TIR version 1 (V1) Level 2 (L2) product for the year 2010 through comparisons with UTLS $CO_2$ data collected with broad spatial coverage by Continuous $CO_2$ Measuring Equipment (CME) in the Comprehensive Observation Network for Trace gases by AIrLiner (CONTRAIL) project. In this study, we validated the TIR V1 $CO_2$ product in the lower troposphere (LT) and the middle troposphere (MT) (736−287 hPa) by comparing them with CONTRAIL CME $CO_2$ profiles over airports, and calculated bias-correction values

for the TIR $CO_2$ data, based on comparisons by latitude, pressure layer, and season from 2010 to 2012. We then examined the validity of the bias-correction values evaluated in limited areas over airports by comparing TIR $CO_2$ data before and after applying the bias-correction values to $CO_2$ data simulated using Nonhydrostatic ICosahedral Atmospheric Model (NICAM)-based transport model (TM) (Niwa et al., 2011b).

## 2. GOSAT/TANSO-FTS and CONTRAIL CME observations

GOSAT, launched on 23 January 2009, and has continued operational measurements of $CO_2$ and $CH_4$ for approximately eight years. TANSO-FTS on board GOSAT consists of three bands in the SWIR region and one in the TIR region (Kuze et al., 2009). We analyzed the latest $CO_2$ product from the TIR band of TANSO-FTS, the TIR V1 L2 $CO_2$ product. The TIR V1 L2 $CO_2$ product was generated from TANSO-FTS version 161.160 (V161) Level 1B (L1B) radiance spectra. Saitoh et al. (2016) described the retrieval algorithm for the TIR V1 L2 $CO_2$ product in detail. In the TIR V1 L2 algorithm, $CO_2$

concentrations are retrieved in 28 vertical grid layers from the surface to 0.1 hPa. Saitoh et al. (2016) and Saitoh et al. (2017) evaluated biases in TIR V1 $CO_2$ data in the UTLS region (287−162 hPa) and calculated growth rates and amplitudes of seasonal variations in TIR V1 UT $CO_2$ data. These studies showed: 1) TIR UT $CO_2$ data agreed with CME $CO_2$ data to within 0.1% and an average of 0.5% in the Southern and Northern Hemispheres, respectively; 2) these data exhibited negative biases larger than 2 ppm in spring and summer in northern low and middle latitudes; 3) their negative biases

increased over time partly due to constraint by a priori data with low growth rates taken from National Institute for Environmental Studies (NIES) transport model, NIES-TM05 (Saeki et al., 2013b); and 4) they displayed more realistic seasonal variations in UT $CO_2$ concentrations than a priori data. In this study, we validated the quality of TIR V1 $CO_2$ data in the LT (736−541 hPa) and MT (541−287 hPa) regions, by comparing them to CONTRAIL CME $CO_2$ data. Table 1 shows pressure levels of retrieval grid layers of the TIR V1 $CO_2$ product that this study focused on.

CONTRAIL is a project to observe atmospheric trace gases, such as $CO_2$ and $CH_4$, using two types of instruments installed on commercial aircraft operated by Japan Airlines (JAL) starting in 2005. Of the two instruments, CME can observe $CO_2$ concentrations more frequently over a wide area (Machida et al., 2008). See Machida et al. (2008) and Machida et al. (2011)



for details about CME $CO_2$ observations. This study used $CO_2$ data obtained with CME during the ascent and descent flights over several airports from 2010 to 2012. Figure 1 shows the locations of the airports used here, which fall in the latitude range of 40°S to 60°N.

## 3. NICAM-TM $CO_2$ data

We used atmospheric $CO_2$ data simulated by NICAM-TM (Niwa et al., 2011b) for global comparison with TANSO-FTS TIR $CO_2$ data. NICAM has quasi-homogeneous grids, with horizontal grids generated by recursively dividing an icosahedron. The NICAM simulations used in this study were performed with a horizontal resolution of around 240 km, which corresponds to the horizontal resolution when an icosahedron is divided five times ("glevel-5"). See Tomita and Satoh (2004) and Satoh et al. (2008, 2014) for details of NICAM. The transport model version of NICAM, NICAM-TM, has been

developed and used for atmospheric transport and source/sink inversion studies of long lived species such as $CO_2$ (Niwa et al., 2011a,b, 2012, 2017).

The NICAM-TM $CO_2$ data used here incorporated CONTRAIL $CO_2$ data into the inverse model, in addition to surface $CO_2$ data (Niwa et al., 2012). CONTRAIL $CO_2$ data obtained during ascending, descending, and cruise level flights were categorized into four vertical bins: 575–625, 475–525, 375–425, and 225–275 hPa. The binned CONTRAIL $CO_2$ data were

then incorporated into NICAM-TM inversion calculations to estimate surface $CO_2$ fluxes. Niwa et al. (2012) showed that incorporating the CONTRAIL $CO_2$ data into the surface flux inversion model improved $CO_2$ concentration simulation compared with a simulation using surface $CO_2$ data only. They also demonstrated that the simulated $CO_2$ concentrations based on CONTRAIL $CO_2$ data showed better agreement with independent upper atmospheric $CO_2$ data obtained in the Civil Aircraft for the Regular Investigation of the atmosphere Based on an Instrument Container (CARIBIC) project

(Brenninkmeijer et al., 2007).

## 4. Methods

### 4.1 Bias assessment of TIR $CO_2$ data using CME observations

Vertical distribution of $CO_2$ concentrations can be obtained by CME during the ascent flights from departure airports and the descent flights to destination airports. Figure 2 shows the flight tracks of CME ascending and descending observations over

Narita airport (35.8°N, 140.4°E) in 2010. CME $CO_2$ data were regarded as part of $CO_2$ vertical profiles, with maximum altitudes around 12 km, and were obtained within 3−4° of latitude and longitude of the airport. Therefore, we set the threshold for selecting coincident pairs of TANSO-FTS TIR and CME $CO_2$ profiles for comparison to be a 300-km distance from each of the airports shown in Figure 1.

For each of the coincident pairs, we calculated the weighted average of discrete CME $CO_2$ data in a vertical layer,

"CME_raw", represented by black circles in Figure 3(a), with respect to the center pressure levels of each of the 28 vertical



grid layers of TIR $CO_2$ data. When there were no corresponding CME $CO_2$ data in lower retrieval grid layers, $CO_2$ concentration at the lowest altitude observed by CME was assumed to be constant down to the lowest retrieval grid layer. Similarly, the uppermost $CO_2$ concentration observed was assumed to be constant up to the center pressure level of the retrieval grid layer including the tropopause, identified based on temperature lapse rates of Global Spectral Model Grid Point

Values from the Japan Meteorological Agency (JMA-GPV) interpolated to the location of CME measurement. In retrieval grid layers above the tropopause, $CO_2$ concentrations were determined based on $CO_2$ concentration gradients calculated from NICAM-TM $CO_2$ data near a CME measurement location. We collected eight NICAM-TM $CO_2$ data points from four model grids adjacent to a CME measurement location at times before and after CME measurement, and linearly interpolated them to the CME measurement location and time. The red line in Figure 3(a) shows a $CO_2$ vertical profile determined in this

manner. This $CO_2$ vertical profile was designated as "CME_obs." profile. Observations by satellite-borne nadir-viewing sensors like TANSO-FTS have much lower vertical resolution than aircraft observations. Therefore, we smoothed the CME_obs. profile to fit its vertical resolution to the vertical resolution of corresponding TIR $CO_2$ profile by applying TIR $CO_2$ averaging kernel functions (AK) to the CME_obs. profile, as follows (Rodgers and Connor, 2003):

$$\mathbf{x}_{\mathbf{CME\_AK}} = \mathbf{x}_{\mathbf{a\ priori}} + \mathbf{A}\big(\mathbf{x}_{\mathbf{CME\_obs.}} - \mathbf{x}_{\mathbf{a\ priori}}\big). \qquad (1)$$

Here, $\mathbf{x}_{\mathbf{CME\_obs.}}$ and $\mathbf{x}_{\mathbf{a\ priori}}$ are the CME_obs. and a priori $CO_2$ profiles, respectively. CME_obs. data with TIR $CO_2$ averaging kernels was designated as "CME_AK", as indicated by the blue line in Figure 3(a).

We set two different criteria for the time difference between TANSO-FTS TIR and CME $CO_2$ profiles used for selection of coincident pairs: a 24-h difference and a 72-h difference. Figure 4 shows a comparison of the results over Narita airport for coincident pairs with a 24- or 72-h time difference. Both averages and 1-σ standard deviations of differences between TIR

and CME $CO_2$ data selected using the 24- and 72-h thresholds were comparable, as shown in Figure 4, which means that the use of these two time difference criteria does not alter any conclusions drawn from comparisons of TIR and CME $CO_2$ data. The same was generally applied to comparisons over the other airports shown in Figure 1. Hence, we adopted a 72-h time difference between TIR and CME $CO_2$ measurement times for selecting coincident pairs to increase the number of pairs available.

We selected coincident pairs of TIR and CME_AK $CO_2$ profiles by applying the thresholds of a 300-km distance and a 72-h time difference and calculated the difference in $CO_2$ concentrations (TIR minus CME_AK) for each retrieval grid layer. All the airports we used were then divided into four latitude bands (40°S−20°S, 20°S−20°N, 20°N−40°N, and 40°N−60°N), and average differences were calculated for each latitude band, retrieval layer, and season (northern spring, MAM; northern summer, JJA; northern fall,  SON; and northern winter, DJF). The signs of the calculated average differences were flipped

and defined as "bias-correction values" for the 28 retrieval grid layers, four latitude bands, and four seasons. The comparison area for low latitudes was extended to a band of 20°S−20°N, because the number of coincident pairs in that region was smaller than in other latitude bands. The number of coincident pairs was smallest at 20°S−0°; no data were collected there



after September 2010. Thus, all bias-correction values for 20°S−20°N after the SON season of 2010 were determined based on data from 0°−20°N.

## 4.2 Comparison of TIR $CO_2$ data with NICAM-TM $CO_2$ data

In this study, we compared monthly averaged TANSO-FTS TIR and NICAM-TM $CO_2$ data. We used 2.5° grid data from NICAM-TM glevel-5 $CO_2$ simulations, and calculated monthly averaged TIR and NICAM-TM $CO_2$ data for each of these 2.5° grids. Here, we interpolated the NICAM-TM $CO_2$ data from 40 vertical levels into $CO_2$ concentrations at the 28 retrieval grid layers of TIR $CO_2$ data. Besides TIR $CO_2$ data, a priori $CO_2$ data and TIR $CO_2$ averaging kernel functions data were also averaged for each month and each 2.5° grid. For each of the 2.5° grids, we applied the monthly averaged TIR $CO_2$ averaging kernel functions to the corresponding monthly averaged NICAM-TM $CO_2$ profiles using expression (1) with the corresponding monthly averaged a priori $CO_2$ profiles. We then calculated differences in $CO_2$ concentrations between monthly averaged TIR data and monthly averaged NICAM-TM data with TIR averaging kernel functions for each grid. Here, two types of differences were calculated between TIR $CO_2$ data and NICAM-TM $CO_2$ data with TIR $CO_2$ averaging kernel functions: (1) the difference with respect to the original TIR $CO_2$ data and (2) the difference with respect to bias-corrected TIR $CO_2$ data to which the bias-correction values described above were applied.

TIR $CO_2$ averaging kernel functions depend on TIR measurement spectral noise, a priori $CO_2$ profile variability, and $CO_2$ Jacobians. Of these three parameters, covariance matrices of the TIR measurement noise and a priori $CO_2$ profile were set in the same manner for all TIR V1 L2 $CO_2$ data (Saitoh et al., 2016). The $CO_2$ Jacobians depend on temperature and $CO_2$ profiles, and therefore change with location and time. However, TIR $CO_2$ averaging kernel functions showed nearly identical structures with each other when collected for each 2.5° grid in one month, which means that applying the monthly averaged TIR $CO_2$ averaging kernel functions did not affect the conclusions of this study.

## 5. Results

### 5.1 Bias of TIR LT and MT $CO_2$ concentrations

Figure 5 presents a comparison between TANSO-FTS TIR V1 and CME_AK $CO_2$ profiles over Narita airport in each season in 2010. In all seasons, TIR $CO_2$ data in the LT and MT regions had negative biases against CME_AK $CO_2$ data. The largest negative biases in TIR $CO_2$ data were found in the MT region centered at 500−400 hPa. The peak of the negative biases in spring and summer occurred at ~400 hPa, slightly higher than the peak pressure level in fall and winter (~500 hPa), which corresponds to the pressure level at which the TIR $CO_2$ averaging kernels exhibited their highest sensitivity in each season. Saitoh et al. (2016) showed that TIR V1 $CO_2$ data agreed well with CME level flight $CO_2$ data in the UT region (287−196 hPa). As indicated by the thick black lines in Figure 5, the negative biases in TIR $CO_2$ data against CME ascending and descending flight $CO_2$ data decreased as altitude increased, which is consistent with the results of Saitoh et al. (2016).



Figure 6 shows differences between TANSO-FTS TIR V1 and CME_AK $CO_2$ data in the LT and MT regions for each latitude band and each season. TIR $CO_2$ data had consistent negative biases of 1−1.5% against CME_AK $CO_2$ data in all retrieval layers from 736 to 287 hPa, with the largest negative biases at 541−398 hPa (retrieval layers 5−6) for all latitude bands and seasons, except for 40°S−20°S in the DJF seasons of 2011 and 2012. Here, we have omitted a detailed discussion of TIR $CO_2$ data at pressure levels below 736 hPa (retrieval layers 1−2), because TIR measurements have relatively low sensitivity to $CO_2$ concentrations in these layers, as shown in Figure 3(b). The largest negative biases, up to 7.3 ppm, existed in low latitudes during the JJA season, as indicated by the red line in the upper panel of Figure 6(b), while there were no coincident pairs of TIR and CME $CO_2$ data in the same season of 2011 and 2012. The negative biases in TIR $CO_2$ data were larger in spring and summer than in fall and winter in northern middle latitudes, as was the case for UT comparisons presented in Saitoh et al. (2016). On a global scale, the seasonality of negative biases was not clear, given the relatively large 1-σ standard deviations, although these biases tended to be larger in the spring hemisphere than in the fall hemisphere. Comparing results among the three years, the negative biases in TIR $CO_2$ data slightly increased over time, but not as sharply as in the UT $CO_2$ comparisons discussed in Saitoh et al. (2017). Note that the number of comparison pairs used in Figure 6 varied among latitude bands; the largest number occurred at 20°N−40°N, which includes Narita airport, Japan, and the number of coincident profiles decreased in low latitudes and the Southern Hemisphere, where there are fewer airports.

## 5.2 Validity of bias correction based on CME data

Negative biases in TANSO-FTS TIR V1 $CO_2$ data in the LT and MT regions did not exhibit evident dependence on season or year, as shown in Figure 6. However, it is difficult to discern whether bias assessment using TIR $CO_2$ data over airports reflects the typical features of each latitude band due to the limited airport locations. Therefore, we validated the applicability of the bias-correction values based on comparisons with CME_AK $CO_2$ data over the entire area of each latitude band by comparing TIR $CO_2$ data to NICAM-TM $CO_2$ data to which TIR $CO_2$ averaging kernel functions were applied on a global scale. Figure 7 shows the frequency distributions of differences in monthly averaged $CO_2$ concentrations between TIR and NICAM-TM $CO_2$ data in all retrieval layers from 736 to 287 hPa in all 2.5° grids over the latitude range of 40°S to 60°N. As shown by the dashed lines in Figure 7, the mode values of the frequency distributions generally corresponded to the median values, indicating that TIR $CO_2$ data did not have locally distorted biases against NICAM-TM $CO_2$ data. In addition, negative biases of TIR $CO_2$ data against NICAM-TM $CO_2$ data in all seasons slightly increased over time, judging from the mode values, although the increase in negative biases was not also evident as in the comparisons over airports shown in Figure 6.

The thick lines in Figure 7 show frequency distributions of differences between NICAM-TM $CO_2$ data and bias-corrected TIR $CO_2$ data to which the bias-correction values defined for each retrieval layer, latitude band, and season were applied. The mode values, which were nearly identical to the median values, were closer to zero in all three years. In addition, variability in the differences, as indicated by the width of the distribution, between bias-corrected TIR and NICAM-TM $CO_2$ data was comparable to or smaller than that between the original TIR and NICAM-TM $CO_2$ data. This demonstrates the





validity of the 288 bias-correction values defined for six retrieval layers from 736 to 287 hPa, four latitude bands (0°S−20°S, 20°S−20°N, 20°N−40°N, and 40°N−60°N), and four seasons of 2010−2012. We thus conclude that the bias-correction values defined based on comparisons in limited areas near airports are generally applicable to TIR $CO_2$ data in areas other than the airport locations. However, there were some exceptions during the JJA season. As indicated by the thick black line in Figure 7(c), the frequency distribution of differences between bias-corrected TIR and NICAM-TM $CO_2$ data in the JJA season of 2010 had a clear bimodal feature, with one of the mode values located near 4 ppm.

We divided the frequency distribution in the JJA season of 2010 into three categories based on the retrieval layers: 736−541 hPa (retrieval layers 3−4), 541−398 hPa (retrieval layers 5−6), and 398−287 hPa (retrieval layers 7−8), as shown in Figure 8. A frequency distribution with a mode of 4 ppm was obtained from bias-corrected TIR $CO_2$ data in the MT region above 541 hPa, especially on 398−287 hPa. That is, TIR $CO_2$ data on 398−287 hPa in the JJA season of 2010 were clearly overcorrected when applying the bias-correction values defined in this study. In the retrieval layers of 736−541 hPa, the mode value of the frequency distribution after bias-correction was close to zero and the width of the distribution narrowed, demonstrating the validity of the corresponding bias-correction value. For the JJA seasons of 2011 and 2012, bias-correction values could not be determined because there were no coincident pairs between TIR and CME $CO_2$ data over airports; therefore, we substituted the bias-correction value for the same season of 2010. The frequency distribution of the differences between NICAM-TM and TIR $CO_2$ data after bias-correction in the JJA season of 2011 had a somewhat bimodal shape, while that in the JJA season of 2012 did not have any bimodal structure, as shown in Figure 7(c). The negative bias of the original TIR $CO_2$ data against NICAM-TM $CO_2$ data in the JJA season of 2012 was larger than that in the JJA season of 2010; thus, applying the bias-correction value for 2010 to the 2012 TIR $CO_2$ data did not lead to any evident overcorrection.

Next, we divided the frequency distribution in the retrieval layers of 398−287 hPa in the JJA season of 2010, shown in Figure 8, into four latitude bands. Judging from the results presented in Figure 9, overcorrection of the negative biases in TIR $CO_2$ data against NICAM-TM $CO_2$ data occurred at 20°S−20°N and 40°N−60°N; TIR $CO_2$ data were markedly overcorrected by the bias-correction value based on comparisons of CME $CO_2$ data over airports, especially in the latitude band of 20°S−20°N. As shown in the upper panel of Figure 6, negative biases in TIR $CO_2$ data against CME $CO_2$ data over airports in low latitudes during the JJA season were clearly larger than the biases found in other latitudes and seasons. Judging from comparisons of global NICAM-TM $CO_2$ data, however, applying bias-correction values based on the negative biases observed over airports to TIR $CO_2$ data over the entire area of 20°S−20°N led to overcorrections in most cases.

## 6. Discussion

Any uncertainties in a priori data can affect retrieval results. A priori $CO_2$ data taken from the NIES-TM05 model (Saeki et al., 2013b) was used in the TANSO-FTS TIR V1 $CO_2$ retrieval processing, and exhibited consistent negative biases against CME $CO_2$ data in the troposphere and the lower stratosphere. As discussed in Saitoh et al. (2016), the negative biases in a priori $CO_2$ data were one likely reason for negative biases in retrieved $CO_2$ concentrations in the UTLS region. The same



pattern holds for negative biases in TIR $CO_2$ data in the LT and MT regions. However, negative biases in retrieved TIR $CO_2$ data were larger than those of a priori $CO_2$ data in the LT and MT regions, as shown in Figure 5. Furthermore, the vertical and latitudinal structures of the negative biases in TIR $CO_2$ data did not always correspond to those in a priori $CO_2$ data. Although negative biases in a priori $CO_2$ data surely contribute to negative biases in TIR V1 $CO_2$ data in the LT and MT

regions, there are likely other considerable sources of TIR $CO_2$ negative biases.

Uncertainty in atmospheric temperature data could affect $CO_2$ retrievals. As shown in Figure 7(a) of Saitoh et al. (2009), uncertainties in retrieved $CO_2$ concentrations due to uncertainties in atmospheric temperature were largest in the UT, upper MT, and LT regions; a bias of 1 K in atmospheric temperature can yield up to ~10% uncertainty in retrieved $CO_2$ concentrations in the MT and LT regions. However, simultaneous retrieval of atmospheric temperature in the V1 $CO_2$

retrieval algorithm could decrease the effect on $CO_2$ retrieval results. In addition to that, no evidence has been reported that the JMA-GPV temperature data used as initial values (equal to a priori values) in the TIR V1 $CO_2$ retrieval processing have biases over such wide latitudinal areas, as in this study. Thus, uncertainty in atmospheric temperature is not a primary cause of negative biases in TIR $CO_2$ data in the LT and MT regions.

As shown in Figure 6, the largest negative biases in TIR V1 $CO_2$ data existed in the MT region in low latitudes (20°S−20°N)

during the JJA season. Degrees of freedom (DF) of TIR V1 $CO_2$ data were highest in low latitudes, exceeding 2.2 in all seasons, which means retrieved $CO_2$ concentrations there contained more information coming from TANSO-FTS TIR L1B spectra and thus were relatively less constrained to a priori concentrations. Kataoka et al. (2014) reported biases in TANSO-FTS TIR V130.131 L1B radiance spectra, which were a previous version of the V161 L1B data used in TIR V1 L2 $CO_2$ retrieval, on the basis of a double difference method. Similar analysis for the V161 L1B spectra is in progress. Kuze et al.

(2016) summarized updates in the processing method for TANSO-FTS L1B spectra and showed that the V161 and newer version (V201) of TANSO-FTS L1B spectra still had considerable uncertainties via theoretical simulations. Kataoka et al. (2014) and Kuze et al. (2016) demonstrated that TANSO-FTS TIR L1B spectra had considerable radiance biases, which were largest at around 15-μm $CO_2$ absorption band.

In the TIR V1 $CO_2$ retrieval algorithm, we simultaneously retrieved surface temperature and surface emissivity with $CO_2$

concentration as a correction parameter for radiance biases in the V161 spectra, as explained in Saitoh et al. (2016). In the $CO_2$ retrieval, these surface parameters were retrieved to correct the radiance biases separately in the three spectral regions of the 15-μm (690−715 cm$^{-1}$, 715−750 cm$^{-1}$, and 790−795 cm$^{-1}$), 10-μm (930−990 cm$^{-1}$), and 9-μm bands (1040−1090 cm$^{-1}$). As reported in Saitoh et al. (2016), the simultaneous retrieval of surface parameters for correction of radiance biases increased the number of normally retrieved $CO_2$ data (by roughly 1.5 times over Narita airport). This demonstrates a certain level of

validity for the correction of radiance biases through simultaneous retrieval of surface parameters for the V161 spectra. However, we note that retrieving surface parameters for radiance bias correction at each wavelength band may affect retrieved $CO_2$ concentrations, and remaining radiance biases after correction at each wavelength band may also affect retrieved $CO_2$ concentrations.



To examine the effect of the simultaneous retrieval of surface parameters at each of the three wavelength bands on retrieved $CO_2$ concentrations, we performed test retrievals of $CO_2$ concentrations using V161 spectra in four cases: using all three of these bands, in the same manner as the V1 algorithm; using two bands, 15-µm and 10-µm; using two bands, 15-µm and 9-µm; and using the 15-µm band only. Figure 10 shows the $CO_2$ retrieval results for two TANSO-FTS observations over Narita

airport in April 2010. As shown in Figure 10(a), negative biases in TIR $CO_2$ concentrations against nearby CME $CO_2$ concentrations in the LT and MT regions became notably smaller when using the 15-µm and 9-µm bands (black dashed line) and the 15-µm band only (black dashed-dotted line), both conditions that did not use the 10-µm band. It is clear that using the 9-µm band did not contribute to negative biases in retrieved $CO_2$ concentrations, judging from the minor difference in $CO_2$ concentrations between the use of all three bands (thick line) and the use of the 15-µm and 10-µm bands (dotted line). In

addition, there were no major differences in retrieved $CO_2$ concentrations among the four retrieval cases when the original V1 $CO_2$ profile did not have distinct negative biases, as shown in Figure 10(b). According to theoretical calculations shown in Figure 13 in Kuze et al. (2016), there were no distinct radiance biases in the 10-µm band in the latest version of the TANSO-FTS TIR spectra. If it is true for observed TIR radiances, our test retrievals imply that simultaneous retrieval of surface parameters for TIR spectra at the 10-µm band with less radiance bias worsened $CO_2$ retrieval results. The test

retrieval results demonstrate that using the 10-µm band in conjunction with the 15-µm and 9-µm bands in the V1 retrieval algorithm is a probable cause of the negative biases in retrieved $CO_2$ concentrations in the LT and MT regions, although this cannot fully explain the biases.

$CO_2$ absorption at 15 µm is considerably larger than that at 9 or 10 µm. However, measurements in the 9-µm and 10-µm bands are most sensitive to $CO_2$ concentrations in the LT and MT regions; the peak sensitivity of the 9-µm and 10-µm bands

occurred on 736−541 hPa and 541−398 hPa, respectively, judging from $CO_2$ Jacobian values. Therefore, using the 9-µm and 10-µm bands in conjunction with the 15-µm band should be useful for retrieving $CO_2$ vertical profiles. In fact, in the case of the retrieval result shown in Figure 10(a), the degree of freedom of $CO_2$ retrieval was 1.93 when using the 15-µm band only, and it increased to 1.94, 1.95, and 1.96 when adding the 9-µm band, the 10-µm band, and both the 9-µm and 10-µm bands, respectively. In the next update of the $CO_2$ retrieval algorithm for TANSO-FTS TIR spectra, we should consider an

improved method for correcting radiance biases in $CO_2$ retrieval processing or adopting the correction of TIR L1B spectra themselves proposed by Kuze et al. (2016).

Bias-correction values determined based on comparisons of CME $CO_2$ data over airports overcorrected negative biases in TIR $CO_2$ data in the upper MT region from 398 to 287 hPa in low latitudes (20°S−20°N) during the JJA season, as shown in Figure 9. The CME data that determined the bias-correction values of the 20°S−20°N latitude band were concentrated in

East Asia, as illustrated in Figure 1. In addition, the bias-correction values for the 20°S−20°N latitude band after the SON season of 2010 were determined from comparisons of CME data at 0°−20°N, because no data were collected at 20°S−0° after September 2010, as mentioned above. Figure 11 shows differences between TIR $CO_2$ data with no bias correction and NICAM-TM $CO_2$ data with TIR $CO_2$ averaging kernel functions on 682 hPa and 314 hPa in July 2010. As shown in the lower panel of Figure 11, TIR $CO_2$ data on 314 hPa had negative biases against NICAM-TM $CO_2$ data in most areas at





$0°−20°N$, and the negative biases were largest near airport locations in East Asia. At $20°S−0°$, on the other hand, TIR $CO_2$ data on 314 hPa were closer to NICAM-TM $CO_2$ data than at $0°−20°N$. Relying on NICAM-TM $CO_2$ data, which incorporates CONTRAIL $CO_2$ data in the inversion, application of bias-correction values determined mainly from comparisons of CME $CO_2$ data in the MT region at $0°−20°N$ to TIR $CO_2$ data over the entire area of low latitudes including

5 $20°S−0°$ produced widespread overcorrection.

In general, there are few areas where we can obtain reliable in situ $CO_2$ data for validation analysis. In particular, there are very few in situ $CO_2$ data in the free troposphere where TIR observations are most sensitive, compared to the surface. In low latitudes, there are relatively strong updrafts, and thus there are larger uncertainties among models than in other areas due to differences in the parameterization of vertical transport. Therefore, a priori $CO_2$ concentrations taken from the NIES-TM05

model (Saeki et al., 2013b) probably have larger uncertainties in the MT region in low latitudes. As retrieved TIR $CO_2$ concentrations were to some extent constrained by a priori concentrations, they possibly had more biases attributed to the a priori uncertainties in the MT region in low latitudes. More in-situ $CO_2$ data in the upper atmosphere in low latitudes are needed to validate both satellite data and model results. In addition, there may also be large biases in retrieved $CO_2$ data in local source and sink regions, where model data are more variable depending on the surface flux dataset. In such areas, it is

difficult to determine bias-correction values that can be applicable over a vast area; it is true in the case of $40°N−60°N$. In conclusion, comprehensive validation analysis of satellite data is still needed to evaluate accuracy both in background regions and in regions with high $CO_2$ variability. Reconsideration of the setting of retrieval grid layers is also needed so that measurement information should be included more prominently in TIR $CO_2$ retrieval results.

Overall, the bias-correction values evaluated in each retrieval layer, latitude band, and season (Figure 6) can be applied to

20 corresponding TIR $CO_2$ data, except at $20°S−20°N$ during the JJA seasons of 2011 and 2011, when bias-correction values were not determined due to a lack of coincident CME $CO_2$ data. In these two cases, we recommended applying bias-correction value 0.5 ppm and 1.0 ppm larger than the corresponding bias-correction value for 2010 to TIR $CO_2$ data for 2011 and 2012, respectively, judging from comparison results between the original TIR and NICAM-TM $CO_2$ data.

## 7. Summary

We evaluated biases of the GOSAT/TANSO-FTS TIR V1 L2 $CO_2$ product in the LT and MT regions (736−287 hPa) by comparing the TIR $CO_2$ profiles with coincident CONTRAIL CME $CO_2$ profiles over airports from 2010 to 2012. Coincident criteria for comparisons of a 300-km distance and a 72-h time difference yielded a sufficient number of coincident pairs, except in low latitudes ($20°S−20°N$) during JJA seasons of 2011 and 2012. Comparisons between TIR $CO_2$ profiles and CME $CO_2$ profiles to which TIR $CO_2$ averaging kernel functions were applied showed that the TIR V1 $CO_2$ data

had consistent negative biases of 1−1.5% against CME $CO_2$ data in the LT and MT regions; the negative biases were the largest on 541−398 hPa (retrieval layers 5−6), and were larger in spring and summer than in fall and winter in northern middle latitudes, as is the case in the UT region (287−196 hPa). Our test retrieval simulations showed that using the 10-μm



$CO_2$ absorption band (930−990 cm$^{-1}$), in addition to the 15-μm (690−750 cm$^{-1}$ and 790−795 cm$^{-1}$) and 9-μm (1040−1090 cm$^{-1}$) bands, increased negative biases in retrieved $CO_2$ concentrations in the LT and MT regions, suggesting that simultaneous retrieval of surface parameters for radiance bias correction at the 10-μm band worsened $CO_2$ retrieval results.

We then performed global comparisons between TIR V1 $CO_2$ data and NICAM-TM $CO_2$ data with considering TIR $CO_2$ averaging kernel functions to confirm the validity of the bias assessment over airports. Differences in $CO_2$ concentrations between TIR and NICAM-TM data approached an average of zero after application of the bias-correction values to TIR $CO_2$ data, demonstrating that the bias-correction values evaluated over airports in limited areas are applicable to TIR $CO_2$ data for the entire areas of 40°S–60°N. Note that applying the bias correction value at 20°S–20°N in the upper MT region (398−287 hPa) during the JJA season resulted in overcorrection of TIR $CO_2$ data.

This study presented bias-correction values for the GOSAT/TANSO-FTS TIR V1 L2 $CO_2$ product evaluated in the LT and MT region (736−287 hPa) in each latitude band and each season of 2010−2012. This information should be useful for further analyses, including $CO_2$ surface flux estimation and transport process studies using TIR $CO_2$ data in the free troposphere, and also helpful for evaluating wavelength-dependent radiance biases in TANSO-FTS TIR spectra to improve TIR $CO_2$ retrieval algorithm.

**Data availability**

GOSAT/TANSO-FTS TIR V1 L2 and a priori NIES-TM05 $CO_2$ data and TIR $CO_2$ averaging kernel data are available at http://www.gosat.nies.go.jp/en/. Contact the CONTRAIL project (http://www.cger.nies.go.jp/contrail/index.html) to access CONTRAIL CME $CO_2$ data. Contact Y. Niwa for detailed information on NICAM-TM $CO_2$ simulations. Contact the corresponding author, N. Saitoh, to obtain the table of bias-correction values for TIR V1 L2 $CO_2$ data evaluated in this study.

**Acknowledgment.**

We thank all the members of the GOSAT Science Team and their associates. We are also grateful to the engineers of Japan Airlines, the JAL Foundation, and JAMCO Tokyo for supporting the CONTRAIL project. This study was funded by the Japan Aerospace Exploration Agency (JAXA). This study was performed within the framework of the GOSAT Research Announcement.

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



Table 1. Pressure levels of retrieval grid layers of GOSAT/TANSO-FTS TIR V1 L2 $CO_2$ data focused on in this study.

| Layer level | Pressure level of each layer (hPa) | Lower pressure level (hPa) | Upper pressure level (hPa) |
|---|---|---|---|
| 1 | 927.79 | 1165.91 | 857.70 |
| 2 | 795.08 | 857.70 | 735.64 |
| 3 | 682.10 | 735.64 | 630.96 |
| 4 | 585.63 | 630.96 | 541.17 |
| 5 | 502.47 | 541.17 | 464.16 |
| 6 | 430.97 | 464.16 | 398.11 |
| 7 | 369.64 | 398.11 | 341.45 |
| 8 | 314.23 | 341.45 | 287.30 |
| 9 | 262.10 | 287.30 | 237.14 |
| 10 | 216.36 | 237.14 | 195.73 |



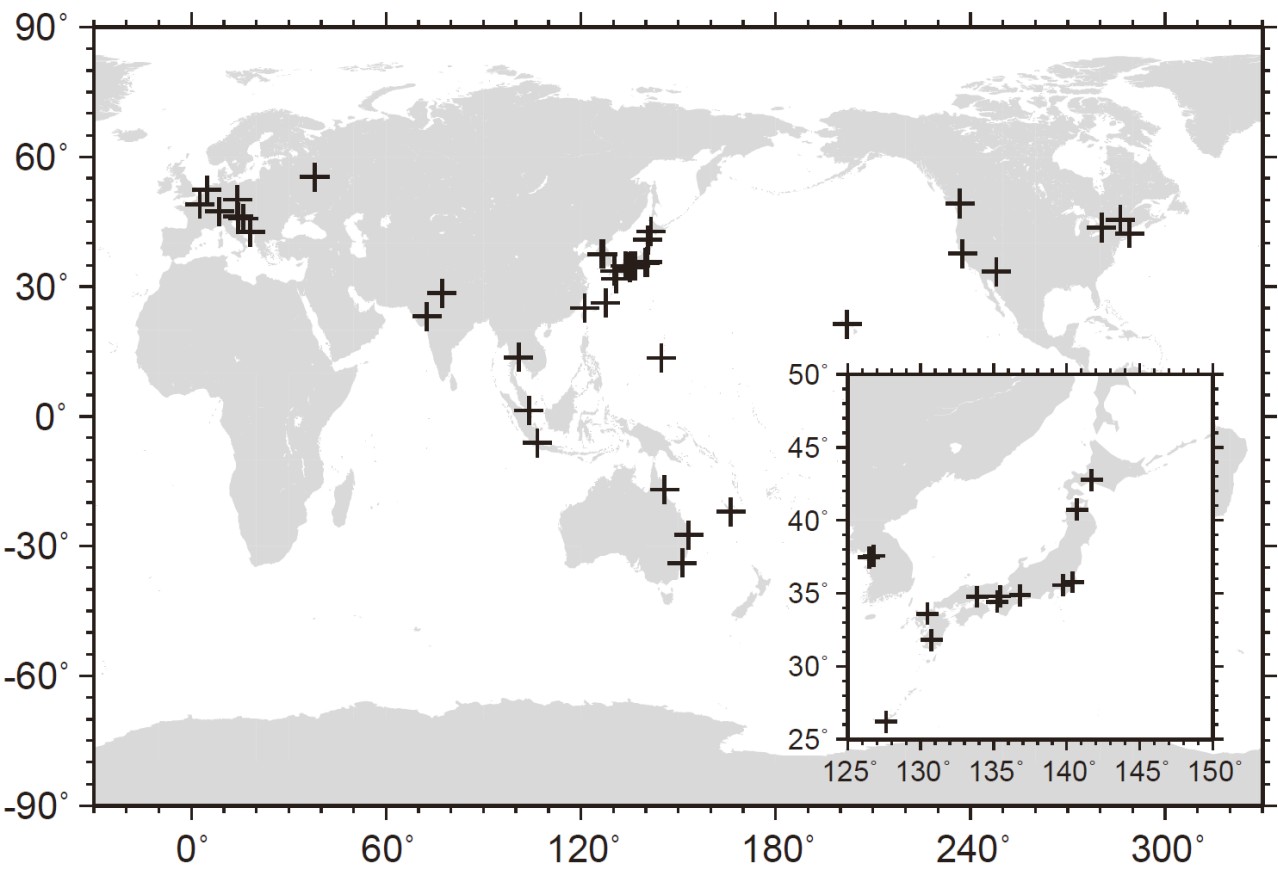

Figure 1. Locations of airports at which CONTRAIL CME ascending and descending observations were collected used in this study.





Figure 2. Flight tracks of all CME ascending and descending observations over Narita airport in 2010. Color indicates the
5    altitude levels of each flight.



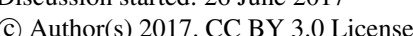


Figure 3. (a) Black circles represent original CME data (CME_raw), the red line shows an interpolated profile of the CME data into GOSAT/TANSO-FTS TIR $CO_2$ 28 retrieval grid layers (CME_obs), the blue line shows the interpolated profile to which TIR averaging kernel functions, shown in panel (b), are applied (CME_AK), and the green line shows a priori $CO_2$ profile.



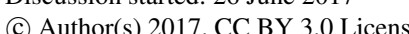

Figure 4. Bias profiles of GOSAT/TASNO-FTS TIR $CO_2$ data against CME_AK $CO_2$ data over Narita airport using
coincident pairs with 24-hour (gray) and 72-hour (black) time difference criteria: (a) winter (JF) 2010 and (b) summer (JJA)
2010.





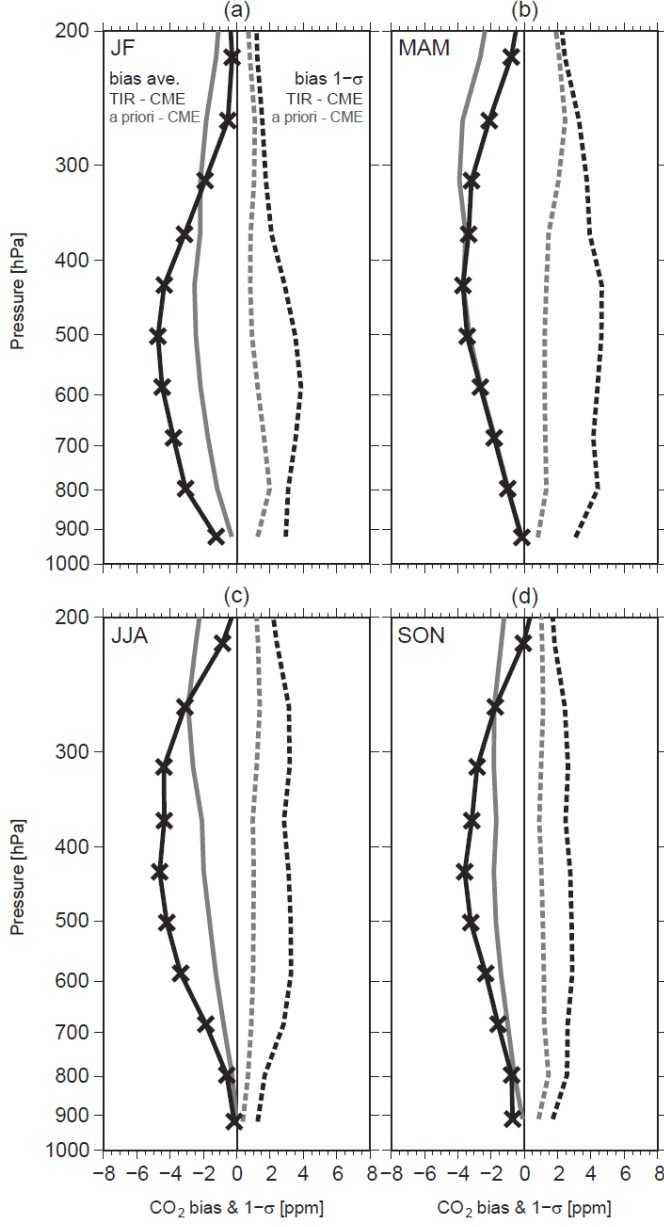

Figure 5. Bias profiles of GOSAT/TANSO-FTS TIR $CO_2$ data and a priori $CO_2$ data against CME_AK $CO_2$ data over Narita
airport and the 1-$\sigma$ standard deviations for each retrieval layer and season in 2010. The CME_AK $CO_2$ data are CME $CO_2$
data to which TIR $CO_2$ averaging kernel functions are applied. Thick black and gray lines indicate the biases of TIR and a
priori $CO_2$ data, respectively, and dotted black and gray lines show their 1-$\sigma$ standard deviations. Cross symbols indicate the
center pressure level of each retrieval layer: (a) JF, (b) MAM, (c) JJA, and (d) SON.




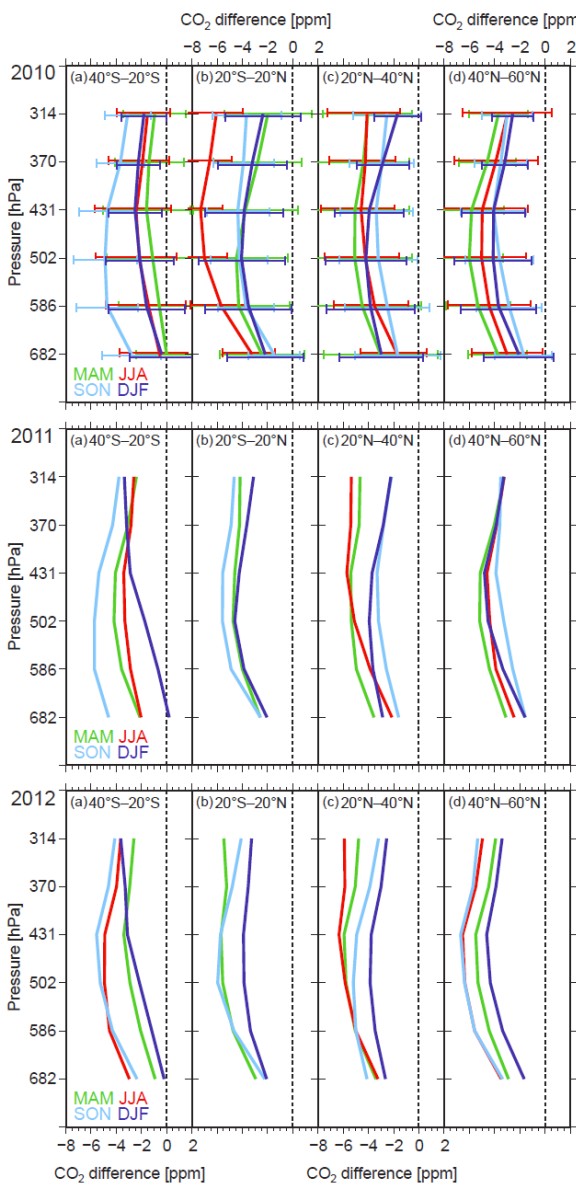

Figure 6. Average differences in $CO_2$ concentrations between GOSAT/TANSO-FTS TIR and CME_AK $CO_2$ data (TIR minus CME_AK) from 736 to 287 hPa (retrieval layers 3–8) for each latitude band and season, 2010–2012. The 1-σ standard deviations of the averages are indicated by horizontal bars for comparison of 2010 as a reference, which are slightly shifted up and down for visibility. We divided the data into four latitude bands: (a) 40°S–20°S, (b) 20°S–20°N, (c) 20°N–40°N, and (d) 40°N–60°N. Green, red, light blue, and blue lines represent the results in northern spring (MAM), northern summer (JJA), northern fall (SON), and northern winter (DJF), respectively.





Figure 7. Frequency distributions of biases of monthly averaged GOSAT/TANSO-FTS TIR $CO_2$ data against monthly averaged NICAM-TM $CO_2$ data evaluated for each of retrieval layers from 736 to 287 hPa for each 2.5° grid in the latitude range of 40°S–60°N. Monthly averaged TIR $CO_2$ averaging kernel functions were applied to NICAM-TM $CO_2$ data in each grid. Thick and dashed lines indicate the biases of the original TIR $CO_2$ data (no bias correction) and bias-corrected TIR $CO_2$ data, respectively. Black, red, and blue lines show results from 2010, 2011, and 2012, respectively.





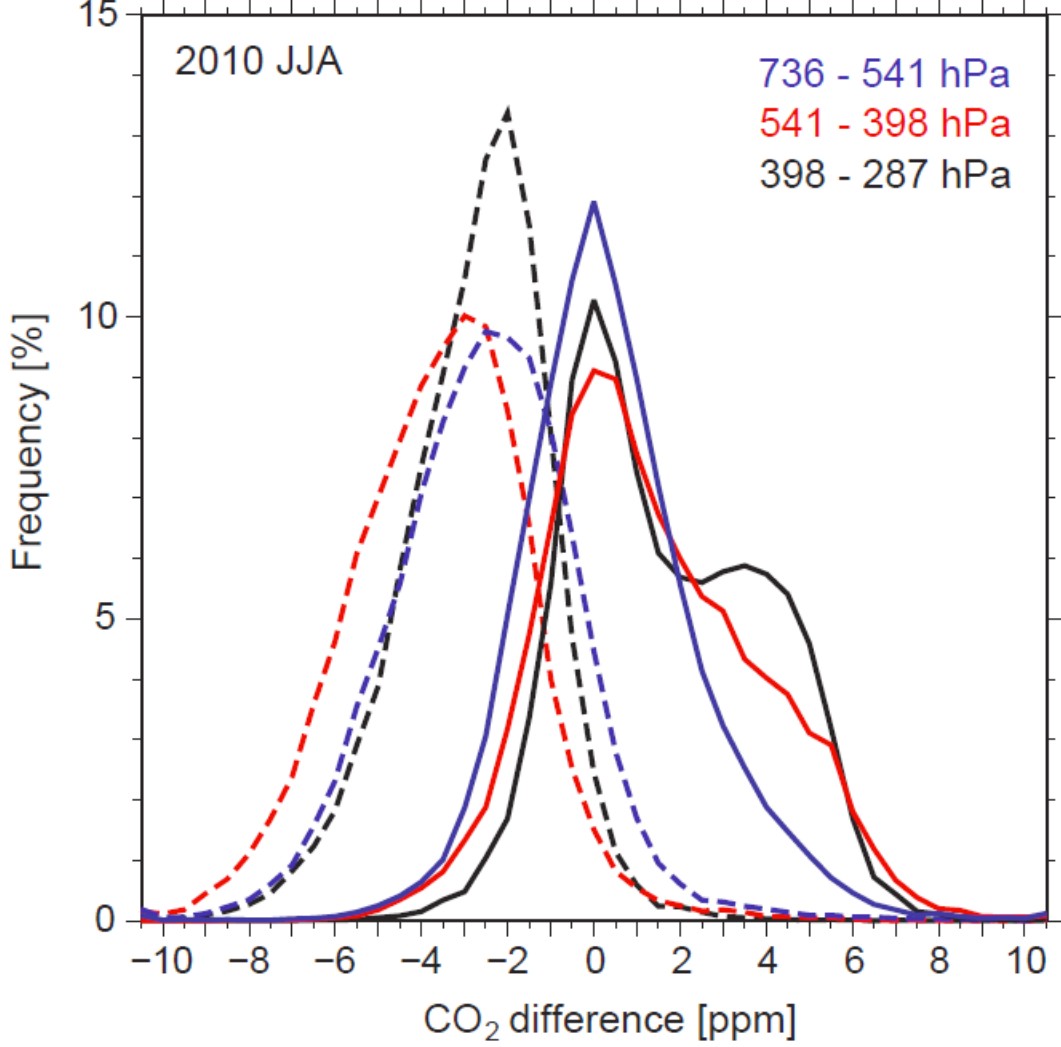

Figure 8. Same as Figure 7, but showing frequency distributions during the JJA season of 2010 on 736–541 hPa (retrieval layers 3–4), 541–398 hPa (retrieval layers 5–6), and 398–287 hPa (retrieval layers 7–8). Black, red, and blue lines indicate the results on 398–287 hPa, 541–398 hPa, and 736–541 hPa, respectively.



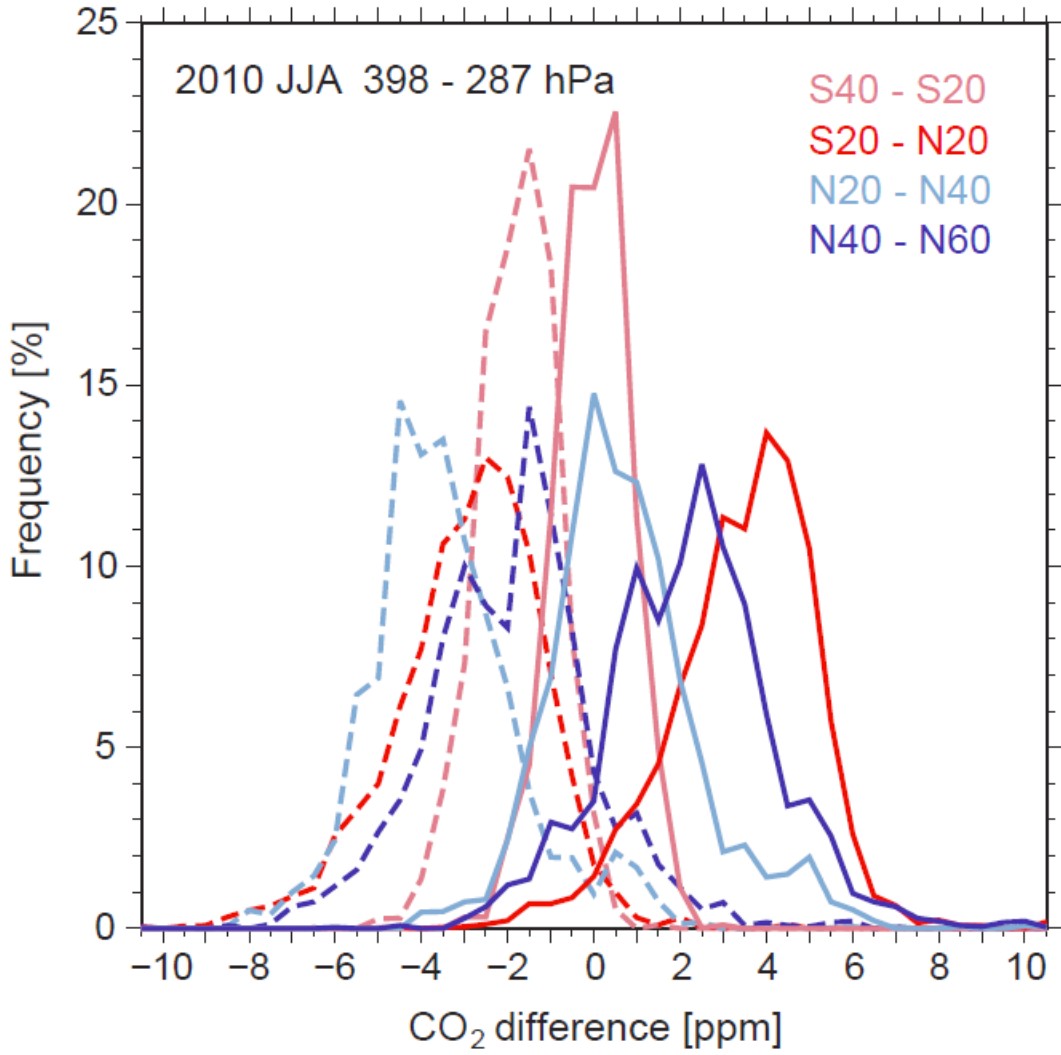

Figure 9. Same as Figure 7, but showing frequency distributions during the JJA season of 2010 on 398−287 hPa (retrieval
5  layers 7–8) for each latitude band. Pink, red, light blue, and blue lines shows the results from 40°S–20°S, 20°S–20°N, 20°N–
40°N, and 40°N–60°N, respectively.



Figure 10. CO$_2$ profiles over Narita airport retrieved using four different wavelength bands of GOSAT/TANSO-FTS V161

5   L1B spectra: three bands, 15-µm, 10-µm, and 9-µm (thick lines); two bands, 15-µm and 10-µm (dotted lines); two bands, 15-µm and 9-µm (dashed lines), and the 15-µm band only (dashed-dotted lines). Nearby CME CO$_2$ profiles (CME_obs.) are shown by gray lines: (a) a case of April 1, 2010 and (b) a case of April 30, 2010.



Figure 11. Latitude–longitude cross-sections of differences in monthly averages of GOSAT/TANSO-FTS TIR $CO_2$ data and
NICAM-TM $CO_2$ data with considering TIR $CO_2$ averaging kernel functions (TIR minus NICAM-TM) in July 2010. The
upper and lower panels show the results on 682 hPa (retrieval layer 3) and 314 hPa (retrieval layer 8), respectively.