# Peer review of "Bias assessment of lower and middle tropospheric CO2 concentrations of GOSAT/TANSO-FTS TIR Version 1 product"

_Atmospheric Measurement Techniques, 2017_

## Referee Comment (RC1) · Anonymous Referee #2 · 27 Jul 2017

Review of Saitoh et al., "Bias assessment of lower and middle tropospheric CO2 concentrations of GOSAT/TANSO-FTS TIR Version 1 product"

Overall, this is a good paper dealing with difficult but necessary bias corrections to TANSO-FTS observations of mid-troposphere CO2. It's a tricky subject, but the methodology is generally sound. However, the paper is difficult to follow in some sections, and in many cases, the figures need some improvement and clarification. I would recommend publication after some revisions in the text, and if the authors could better address the issue of the number of layers in the forward model (see comment for pg 10, line 32 below.)

General comment: Throughout the paper, the authors refer to the retrieval layers by number (layer 3, layer 4, etc.), rather than, say, its log mean pressure. These layer numbers are specific to their algorithm, and referencing the layers by number is a little burdensome to the reader, even where the pressures are provided. For example, Page 6, line 23 reads "Saitoh et al. (2016) showed that TIR V1 CO2 data agreed well with CME level flight CO2 data in the UT region corresponding to retrieval layers 9 and 10." This would read better if the pressures were given instead of the layer numbers. I suggest they prepare a table listing the retrieval layer numbers, layer boundary pressures, and the log-mean pressures of the layers (similar to Table 1 of Saitoh et al., 2016), and then just refer to a layer by its mean pressure rather than its number.

Pg 1, line 14: "...good spatial representability." It's not obvious what 'representability' means here. Would "resolution and precision" be a better phrase to use?

Pg 1, Line 24: "(retrieval layers 5−6), ..." It's not necessary to get into the details of their retrieval method in the abstract.

Pg 2, line 3: Suggest changing "(e.g., Gurney et al., 2002 Gurney et al., 2004)" to "(e.g., Gurney et al., 2002; 2004)".

Line 24: "spatial representability." Again, not obvious what it means here.

Pg 3, line 16: Suggest changing "...and has continued CO2 and CH4 operational measurements for approximately eight years." to "and has continued operational measurements of CO2 and CH4 for approximately eight years.

Line 23: Suggest shortening "These studies showed the following: 1) TIR UT CO2 data agreed..." to "These studies showed: 1) TIR UT CO2 data agreed..." Page 5, line 14: Suggest more explanation of why the averaging kernels are applied to the CME data and then comparison made. This would be useful to the reader not well versed in averaging kernels etc.

Page 6, Section 4.2: It's not obvious why an "average" averaging kernel can be applied

and not sometimes be misleading. In addition to the effect of instrument parameters (SNR, spectral resolution, view angle etc.) and assuming clear scenes only, the averaging kernel could vary by temperature gradient and thermal contrast with the surface. How much does an averaging kernel vary within a grid box? It would help if the authors briefly explain why they're using an averaged AK here and discuss the limitations of doing so.

Page 7, line 14 "In addition, negative biases of TIR $CO_2$ data against NICAM-TM $CO_2$ data increased by 1 ppm or less per year in all seasons, judging from the mode values, although the increase in negative biases was not evident in the comparisons over airports shown in Figure 6." I did not quite understand what is meant by this. Do they mean the bias varied by 1ppm or less?

Pg 8, line 27: Typo: "... in the LT and ML regions." Did they mean "MT" regions?

Pg 9, line 13: "As shown in Figure 6, the largest negative biases in TIR V1 $CO_2$ data existed in the MT region in middle and low latitudes during spring and summer, where TANSO-FTS TIR measurements have relatively large sensitivity to $CO_2$ concentrations and thus the retrievals are less constrained to a priori concentrations." Some kind of comparison is in order to quantify the difference in $CO_2$ sensitivity here – say average row-sum of averaging kernels, or total DOFS as a function of latitude.

Line 15: "This implies that biases in L1B spectra are a major cause of the negative biases in retrieved $CO_2$ concentrations, as Saitoh et al. (2016) noted in the UT region." The wording is confusing. Does this mean there are biases in the L1b radiances related to latitude and season, or are there fitting biases from the retrieval algorithm? Judging from the rest of the paragraph where the authors write about retrieval of surface parameters, I think they're referring to fitting bias, but whatever the bias is, it should be explicitly described.

Page 10, line 4: "From these results, we conclude that using the 10-$\mu$m band in conjunction with the 15-$\mu$m and 9-$\mu$m bands in the V1 retrieval algorithm is a probable

cause of the negative biases in retrieved $CO_2$ concentrations in the LT and MT regions." While I don't disagree with this, this would be more convincing if the authors compared their results using the different mixes of $CO_2$ bands directly against the aircraft measurements.

Line 13: "According to Figure 13 in Kuze et al. (2016), there was no distinct uncertainty in the 10-$\mu$m band in the latest version of the TANSO-FTS TIR spectra." The wording of this leaves me uncertain of what they're claiming. Uncertainty of linestrengths or low fitting residual? Are they saying that using the 10 micron band of $CO_2$ does not add significant bias? This should be clarified.

Paragraph beginning line 17: As noted earlier, it would really help the reader if the authors referred to the retrieval layers by pressure and not layer number.

Line 32: "In retrieval from TIR spectra, the more atmospheric layers in which we retrieve $CO_2$ concentrations, the lower the information content of the retrieval result in each layer becomes; as a result, the retrieved concentrations are constrained by a priori model data. Thus, there is a high possibility of large biases in retrieved TIR $CO_2$ concentrations in low latitudes." This assertion needs to be tested. It is true that with more layers, the information is spread out more, but the overall information content, as measured by the degrees-of-freedom-of-signal (trace of the averaging kernel) can be the same or very similar, as can the retrieved profiles (depending on what the off-diagonals are for the a priori background covariance.) It's quite possible that if the background a priori is biased, then a TIR retrieval can also be biased not because of the number of retrieval layers, but, particularly at low latitudes, because of water vapor interference, undetected boundary-layer clouds changing the thermal contrast with the surface, or biases in the temperature. Again, this needs to be tested, or the statement removed or at least reworded as a hypothesizing.

Figure 5: It would be much clearer to the reader if they provided guidance to the different panels and lines in a legend box on the figure, rather than only in the caption.

It would also help, for a reader skimming the paper, to describe what "CME-AK CO2" means in the caption as well as the text of the paper.

Figure 6: Use pressures and not layer numbers on vertical axis. It would also be better if latitude information and season (line color) were provided as a legend on the figure. It would help if the lines in the top panels had slight vertical offsets to clarify how different the error bars are from each other.

Figure 7: It's not clear here (or in the text) at what pressures they are comparing avg CO2 with NICAM. The contrast between the mid-gray and light-gray lines is not enough to easily distinguish between them.

Figure 8: Please use pressures instead of layer numbers. Again, the contrast between the mid-gray and light-gray lines is not enough to easily distinguish between them.

Figure 9: Again, please state the pressures instead of "layer 7-8."

Figure 10: Please also describe the lines and the location/times the different panels represent as a legend rather than just in the caption.

---

## Referee Comment (RC2) · Anonymous Referee #1 · 17 Aug 2017

The paper assesses biases in satellite-retrieved CO2 concentrations at the lower and middle troposphere from GOSAT/TANSO-FTS TIR V1 product by comparing them with precise aircraft measurements by CONTRAIL CME, followed by global comparisons of bias-corrected CO2 concentrations with model-simulated CO2 by NICAM-TM. The authors found that the TIR data had negative biases of 1-1.5% against the aircraft measurements and bias-corrected TIR data showed generally good agreement with the NICAM-TM CO2 data, which demonstrated the validity of the bias-correction values.

Observational CO2 data in the free troposphere is still limited, and CO2 profiles from high-resolution GOSAT TIR spectra will help to elucidate CO2 variations in the free

troposphere with its global coverage. Bias estimation of satellite-based CO2 products is highly important for data users and further analysis of CO2 fluxes by atmospheric inversion/data-assimilation studies. The paper is generally well written, and I recommend accepting it for publication after the comments listed below have been addressed.

General comments

1. Results section.

The paper presents comparisons between the original TIR data and CONTRAIL CME data and between bias-corrected TIR data and NICAM-TM data. But the expressions of the evaluations are often qualitative, such as "relatively low", "tend to be larger", "slightly increase", "nearly identical", "close to zero" without any supporting numbers. Although one can see tendencies on the plots, I would recommend illustrating the point with some numbers and add a table with quantitative values to explain the results clearly. The authors do not need to write all related numbers, but at least it would be better to write statistic values related to Figure 7, one of the main plots, to show the validity of the bias-correction values quantitatively. Statistic values in a table or the main text may help readers to follow the discussion. They can be mode values (or medians), standard deviations, kurtoses and skewnesses of frequency distributions, the total number of data pairs, or whatever the authors need to describe Figure 7.

2. "East Asia" in abstract and discussion section.

The authors conclude that one of the reasons of the overcorrection in JJA/low latitudes (0S-20N)/upper MT region is that the correction values were determined by using the data over East Asian airports. Since the authors write this finding to the abstract, this conclusion is thought to be important for the paper. But the explanation (p.10, L34 - L11, L8) is not clear enough to understand why data in the East Asia region strongly affects to the 0-20N bias correction. Usually, Asia in 20S-20N is called Southeast Asia (or part of South India). Do the authors mean "Southeast Asia" rather than "East Asia"?

Or if the East Asian data truly affects the 0-20N bias-correction values via atmospheric transport, please give more explanation and references.

Specific comments

p.3, Section 2, TIR data: Does the TIR product include nighttime data as well as day-time data? I suggest writing time of the observations briefly somewhere in this section.

p.4, Section 3, NICA-TM data: NICAM-TM inversion with CONTRAIL data was con-ducted for the period 2006-2008 (Niwa et al., 2012). It should be explained briefly how the 2010-2012 $CO_2$ data was calculated by NICAM-TM.

p.5, L24, "the number of pairs": Could the authors show the number of pairs which finally used for the comparisons for each latitude bands?

p.7, L10, "On a global scale, the seasonality of negative biases was not clear, given the relatively large 1-$\sigma$ standard deviations, although these biases tended to be larger in the spring hemisphere than in the fall hemisphere.": The sentence is not clear. Does this mean the negative biases had measurable spring-fall seasonality, but it was not statistically significant due to the large standard deviations? Or actually, the biases had no seasonality?

p.7, L26, "negative biases of TIR $CO_2$ data against NICAM-TM $CO_2$ data in all seasons slightly increased over time": Is there no possibility that small trend error in NICAM-TM $CO_2$ could attribute the bias increase in Fig.7? The NICAM-TM natural fluxes were es-timated for the period 2006-2008, which is different from the target period of this article. In other words, does the NICAM $CO_2$ have no bias in trends against CONTRAIL CME data? The authors can confirm it by plotting NICAM-TM $CO_2$ data against CONTRAIL CME data like Fig.6.

p.9, L5, other sources of negative biases: I'm not familiar with retrieval algorithms, but would any errors in cloud detection process cause retrieval errors in the low latitudes with enhanced convective activity? And $H_2O$ or $O_3$ do not affect the $CO_2$ retrieval

results?

p.10, L29-30, "The CME data that determined the bias-correction values of the $20°S−20°N$ latitude band were concentrated in East Asia": I was confused with this sentence. Please see my general comment #2.

p.10, L34 – p.11, L1, "in most areas at $0°−20°N$, and the negative biases were largest near airport locations in East Asia.": Same as above. Please see my general comment #2.

p.11, L12-13, "More in-situ $CO_2$ data in the upper atmosphere in low latitudes": Hiaper Pole-to-Pole Observations (HIPPO) project observed latitudinal distributions of $CO_2$ concentrations in the free troposphere over the Pacific Ocean where mostly clean during 2009 to 2011 (e.g. Wofsy et al., 2011). The dataset has been used for transport model or satellite data validation (e.g. Wecht et al., 2012; Kulawik et al., 2013). The comparison with HIPPO data is out of the scope of this paper, but if the authors found some problems in using HIPPO data for validation, please write it in the discussion section or the introduction section.

Wofsy, S. C. et al.: HIAPER Pole-to-Pole Observations (HIPPO): fine- grained, global-scale measurements of climatically important atmospheric gases and aerosols, Phil. Trans. Roy. Soc. A: Math. Phys. Eng. Sci., 369, 2073–2086, doi:10.1098/rsta.2010.0313, 2011.

p.11, L17, "Reconsideration of the setting of retrieval grid layers . . .": Why do the authors think the current setting of retrieval grid layers might not be suitable for retrievals and reconsideration might solve it?

p.11, L20, "during the JJA seasons of 2011 and 2011": Does this mean "2011 and 2012"?

Figs.3: The Y axis is described in altitude, not in pressure as seen in the following plots. For easy reference, I would suggest adding a 2nd Y axis in pressure or adding

a column in Table 1 to show altitude [km] for each pressure levels. (Rough altitudes from International Standard Atmosphere or the same kind might be enough for this purpose.)

Fig.4: Please replace "Altitude [km] in Y axis label with "Pressure [hPa]".

Fig.7: I think drawing zero lines (i.e. no bias) in each panel makes the bias correction validity more visible.

Fig.7 caption "Thick and dashed lines indicate the biases of the original TIR CO2 data (no bias correction) and bias-corrected TIR CO2 data, respectively.":

1. On my screen, all lines in each panel seem to have same line thickness. Do the authors mean "solid and dashed lines"?

2. This sentence does not match the main text which says that thick lines are bias-corrected values.

Fig.11, gray shade: Could the authors explain what gray zones in the figure are? (No data or out of color scale?)
* * *

---

## Author Comment (AC2) · 11 Sep 2017

To Referee #1,

We appreciate you reading our paper carefully and giving valuable comments and suggestions. We have considered your recommendations for revisions and made the necessary changes. The major points that we deal with in the revised manuscript are as follows:

1.    Following your advice, we have added Table 2 to present bias values of GOSAT/TANSO-FTS thermal infrared (TIR) version 1 (V1) Level 2 (L2) CO2 data

[Figure]

against CONTRAIL CME $CO_2$ data to which TIR $CO_2$ averaging kernel functions were applied. It could help readers see Figure 6. 2. Following your advice, we have added Table 3 to present mode values of frequency distributions of differences in monthly averaged $CO_2$ concentrations between original or bias-corrected TIR and NICAM-TM $CO_2$ data and numbers of data categorized into the mode values and all 2.5° gridded data used for comparisons. It could help readers see Figure 7.

Individual responses to the Referee's comments are attached as a supplement PDF file.

Please also note the supplement to this comment:
https://www.atmos-meas-tech-discuss.net/amt-2017-111/amt-2017-111-AC2-
supplement.pdf

—————————————————————————

---

## Author Response (AR1)

**Reply to the comments by Referees**
* * *
**To Associate Editor and Referees,**

We appreciate you reading our paper carefully and giving valuable comments and suggestions again. We have considered your recommendations for revisions and made the necessary changes. The major points that we deal with in the revised manuscript are as follows:

1. Following the advice of the Referee #2, we have added Table 1 to show representative pressure levels of each of the retrieval grid layers of GOSAT/TANSO-FTS thermal infrared (TIR) version 1 (V1) Level 2 (L2) $CO_2$ product. (We have already addressed this in AMTD.)
2. Relating to the above, we have referred to the retrieval grid layers by the representative pressure levels throughout the text. (We have already addressed this in AMTD.)
3. Following the advice of the Referee #1, we have added Table 2 to present bias values of GOSAT/TANSO-FTS TIR V1 L2 $CO_2$ data against CONTRAIL CME $CO_2$ data to which TIR $CO_2$ averaging kernel functions were applied. It could help readers see Figure 6.
4. Following the advice of the Referee #1, we have added Table 3 to present mode values of frequency distributions of differences in monthly averaged $CO_2$ concentrations between original or bias-corrected TIR and NICAM-TM $CO_2$ data and numbers of data categorized into the mode values and all 2.5° gridded data used for comparisons. It could help readers see Figure 7.

Individual responses to the Referees' comments are listed below.

**Reply to Referee #1,**

*The paper assesses biases in satellite-retrieved $CO_2$ concentrations at the lower and middle troposphere from GOSAT/TANSO-FTS TIR V1 product by comparing them with precise aircraft measurements by CONTRAIL CME, followed by global comparisons of bias-corrected $CO_2$ concentrations with model-simulated $CO_2$ by NICAM-TM. The authors found that the TIR data had negative biases of 1-1.5% against the aircraft measurements and bias-corrected TIR data showed generally good agreement with the NICAM-TM $CO_2$ data, which demonstrated the validity of the bias-correction values.*

*Observational $CO_2$ data in the free troposphere is still limited, and $CO_2$ profiles from high-resolution GOSAT TIR spectra will help to elucidate $CO_2$ variations in the free troposphere with its global coverage. Bias estimation of satellite-based $CO_2$ products is highly important for data users and further analysis of $CO_2$ fluxes by atmospheric inversion/data-assimilation studies. The paper is*

*generally well written, and I recommend accepting it for publication after the comments listed below*

*have been addressed.*

*General comments:*

*1. Results section: The paper presents comparisons between the original TIR data and CONTRAIL*

*CME data and between bias-corrected TIR data and NICAM-TM data. But the expressions of the*

*evaluations are often qualitative, such as "relatively low", "tend to be larger", "slightly increase",*

*"nearly identical", "close to zero" without any supporting numbers. Although one can see tendencies*

*on the plots, I would recommend illustrating the point with some numbers and add a table with*

*quantitative values to explain the results clearly. The authors do not need to write all related numbers,*

*but at least it would be better to write statistic values related to Figure 7, one of the main plots, to*

*show the validity of the bias-correction values quantitatively. Statistic values in a table or the main*

*text may help readers to follow the discussion. They can be mode values (or medians), standard*

*deviations, kurtoses and skewnesses of frequency distributions, the total number of data pairs, or*

*whatever the authors need to describe Figure 7.*

Reply:

We totally agree with you. As described above, we have added Table 2 and Table 3 to present specific values of what we focused on in Figure 6 and Figure 7, respectively. In the revised manuscript, we have referred to Table 2 and Table 3 to clarify points of discussions related to

Figure 6 and Figure 7. We have also referred to specific values presented in Table 2 and Table 3

in the main text of the revised manuscript. We appreciate your comment.

*2. "East Asia" in abstract and discussion section: The authors conclude that one of the reasons of the*

*overcorrection in JJA/low latitudes (0S-20N)/upper MT region is that the correction values were*

*determined by using the data over East Asian airports. Since the authors write this finding to the*

*abstract, this conclusion is thought to be important for the paper. But the explanation (p.10, L34 - L11,*

*L8) is not clear enough to understand why data in the East Asia region strongly affects to the 0-20N*

*bias correction. Usually, Asia in 20S-20N is called Southeast Asia (or part of South India). Do the*

*authors mean "Southeast Asia" rather than "East Asia"? Or if the East Asian data truly affects the 0-20N*

*bias-correction values via atmospheric transport, please give more explanation and references.*

Reply:

We greatly appreciate you pointing out this. We wrote "East Asia" incorrectly in the sentences where we should have written "Southeast Asia" in the manuscript. We intended to say that the bias-correction values in low latitudes (20°S−20°N) in the JJA season in 2010 were determined on the basis of comparisons over the three airports over Southeast Asia: BKK (Bangkok), SIN

(Singapore), and CGK (Jakarta). In the revised manuscript, we have replaced "East Asia" with

"Southeast Asia" throughout the text and described these specific airports in the discussion part.

*Specific comments:*

*Page 3, Section 2, TIR data: Does the TIR product include nighttime data as well as daytime data? I suggest writing time of the observations briefly somewhere in this section.*

Reply:

The TIR products of GOSAT/TANSO-FTS include data obtained both in daytime and nighttime. Following your suggestion, we have stated this clearly in the revised manuscript as follows:

"The TIR band of TANSO-FTS makes observations both in daytime and nighttime, unlike the SWIR band."

*Page 4, Section 3, NICAM-TM data: NICAM-TM inversion with CONTRAIL data was conducted for the period 2006-2008 (Niwa et al., 2012). It should be explained briefly how the 2010-2012 $CO_2$ data was calculated by NICAM-TM.*

Reply:

We agree with you. As you pointed out, the NICAM-TM inversion simulation that was conducted in Niwa et al. (2012) used CONTRAIL and surface $CO_2$ data in 2006–2008 to estimate the natural flux of $CO_2$. The NICAM-TM $CO_2$ data used here were generated by using the estimated $CO_2$ natural flux (fixed for 2010−2012) and year-dependent $CO_2$ fluxes from fossil fuel and biomass burning emissions (considering their yearly trends). Following your suggestion, we have added more explanation of the NICAM-TM $CO_2$ inversion as follows:

"In this study, simulation of NICAM-TM used inter-annually varying flux data of fossil fuel emissions (Andres et al., 2013) and biomass burnings (van der Werf et al., 2010), and the residual natural fluxes from the inversion of Niwa et al. (2012), which mostly represent fluxes from the terrestrial biosphere and oceans. The inversion analysis of Niwa et al. (2012) was performed for 2006−2008 and the three-year-mean fluxes were used in this study."

We appreciate your comment.

*Page 5, line 24, "the number of pairs": Could the authors show the number of pairs which finally used for the comparisons for each latitude bands?*

Reply:

Following your suggestion, we have described the numbers of coincident pairs of TIR and CME_AK $CO_2$ profiles for each of the four latitude bands in the fourth paragraph of Chapter 4.1 in the revised manuscript:

"The numbers of coincident pairs of TIR and CME_AK $CO_2$ profiles varied depending on latitude band and season. The largest number of coincident pairs was obtained in the latitude band of 20°N−40°N including Narita airport, where 506−2501 pairs were obtained. 63−310 and 77−472 coincident pairs were obtained at 40°S−20°S and 40°N−60°N, respectively. The comparison area for low latitudes was extended to a band of 20°S−20°N, because the number of coincident pairs in that region was smaller (0−341) than in other latitude bands; nevertheless, there were no coincident pairs at 20°S−20°N in the JJA seasons of 2011 and 2012. The number of coincident pairs was smallest (0−30) at 20°S−0° and no data were collected there after September 2010. Thus, all bias-correction values for 20°S−20°N after the SON season of 2010 were determined based on data from 0°−20°N."

The below-attached table shows the numbers of the coincident pairs for each season for each latitude band.

|  |  | 40ºS−20ºS | | 20ºS−0º/0º−20ºN | | 20ºN−40ºN | | 40ºN−60ºN | |
|---|---|---|---|---|---|---|---|---|---|
| 2010, MAM | 2010, JJA | 63 | 75 | 27/114 | 30/95 | 1305 | 2501 | 472 | 161 |
| 2010, SON | 2010, DJF | 128 | 114 | 0/172 | 6/155 | 2133 | 1588 | 454 | 132 |
| 2011, MAM | 2011, JJA | 209 | 183 | 0/49 | 0/0 | 506 | 1255 | 77 | 227 |
| 2011, SON | 2011, DJF | 179 | 78 | 0/137 | 0/234 | 1529 | 1049 | 199 | 253 |
| 2012, MAM | 2012, JJA | 310 | 105 | 0/49 | 0/0 | 748 | 1815 | 418 | 406 |
| 2012, SON | 2012, DJF | 145 | 166 | 0/31 | 0/341 | 2045 | 1664 | 326 | 119 |

*Page 7, line 10, "On a global scale, the seasonality of negative biases was not clear, given the relatively large 1-σ standard deviations, although these biases tended to be larger in the spring hemisphere than in the fall hemisphere.": The sentence is not clear. Does this mean the negative biases had measurable spring-fall seasonality, but it was not statistically significant due to the large standard deviations? Or actually, the biases had no seasonality?*

Reply:

In northern middle latitudes (20°N−40°N), negative biases in TIR $CO_2$ data were larger in spring (MAM) and summer (JJA) than in fall (SON) and winter (DJF). On a global scale from 40°S−60°N, any statistically significant seasonality was not found in negative biases in TIR $CO_2$ data against CONTRAIL CME_AK $CO_2$ data. In Table 2 of the revised manuscript, we have presented bias values of TIR $CO_2$ data against CME_AK $CO_2$ data in each season at 541−464 hPa and 464−398 hPa (corresponding to layers 5−6) to make readers refer to specific values that we focused on.

*Page 7, line 26, "negative biases of TIR $CO_2$ data against NICAM-TM $CO_2$ data in all seasons slightly increased over time": Is there no possibility that small trend error in NICAM-TM $CO_2$ could attribute the bias increase in Fig.7? The NICAM-TM natural fluxes were estimated for the period 2006-2008, which is different from the target period of this article. In other words, does the NICAM $CO_2$ have no bias in trends against CONTRAIL CME data? The authors can confirm it by plotting NICAM-TM $CO_2$ data against CONTRAIL CME data like Fig.6.*

Reply:

As explained above, the NICAM-TM inversion simulation that was conducted in Niwa et al. (2012) used CONTRAIL and surface $CO_2$ data in 2006–2008 to estimate the natural flux of $CO_2$. When calculating $CO_2$ concentrations in 2010–2012, the mean inversion fluxes were cyclically used, but fossil fuel and biomass burning $CO_2$ fluxes used were varied inter-annually. We confirmed that the growth rate of the calculated NICAM-TM $CO_2$ concentrations for 2010–2012 is reasonable (2.4 ppm/yr) judging from an observation-based growth rate (2.2 ppm/yr), which is partly contributed by the fact that there were no major El Nino events for both the periods. The below-attached figure shows comparison between the NICAM-TM $CO_2$ simulations and observations at the surface station at Minamitorishima, which demonstrates the validity of the NICAM-TM $CO_2$ simulations. As Figure 6 is based on one-by-one coincident GOSAT−CME_AK $CO_2$ comparisons over airports selected by applying the thresholds of a 300-km distance and a 72-h time difference, we think that it is inappropriate to plot comparisons between 2.5°-gridded NICAM-TM and CME $CO_2$ data on the same figure. Alternatively, we have described the specific comparison in $CO_2$ growth rates between NICAM-TM simulation and surface observation data as follows:

"Furthermore, the $CO_2$ forward simulation of NICAM-TM for 2010−2012 showed a good agreement with in-situ $CO_2$ observations not only in seasonal cycles but also in trends in spite of using the fluxes optimized for 2006−2008; the simulated growth rate at the Minamitorishima station (e.g., Wada et al., 2011), which is one of the global stations of the Global Atmospheric Watch (GAW), was 2.4 ppm/yr for 2010−2012, while the growth rate based on in-situ observations was 2.2 ppm/yr."

"In addition, negative biases of TIR $CO_2$ data against NICAM-TM $CO_2$ data in all seasons slightly increased over time, judging from the mode values presented in the top left boxes of Table 3, although the increase in negative biases was not much evident as in the comparisons over airports shown in Figure 6; this may be partly because of slightly high growth rate of NICAM-TM simulations (2.4 ppm/yr) compared to in-situ observations (2.2 ppm/yr)."

We greatly appreciate your comment.

[Figure]

Reference figure. Time-series of observed (black) and simulated (red) $CO_2$ concentrations at the surface station at Minamitorishima. The observation data presented here were taken from the World Data Center for Greenhouse Gases (WDCGG). The observations have been conducted by JMA under the program of WMO/GAW. We would like to acknowledge the staff that supports the observations.

*Page 9, line 5, other sources of negative biases: I'm not familiar with retrieval algorithms, but would any errors in cloud detection process cause retrieval errors in the low latitudes with enhanced convective activity? And $H_2O$ or $O_3$ do not affect the $CO_2$ retrieval results?*

Reply:

We appreciate your comment. As you pointed out, uncertainties in $H_2O$ and $O_3$ data could also affect $CO_2$ retrievals, as shown in Figure 7(b) and (c) of Saitoh et al. (2009). The TIR V1 $CO_2$ retrieval algorithm (Saitoh et al., 2016) simultaneously retrieves $H_2O$ and $O_3$ with $CO_2$, which could decrease the effect of their uncertainties on $CO_2$ retrieval results. However, water vapor is abundant in the tropics, so that we cannot completely deny the possibility of the effect of $H_2O$ uncertainty on $CO_2$ retrieval results. Similarly, error in the judgement of cloud contamination may affect $CO_2$ retrieval results. We have added this point to the discussion part of the revised manuscript as follows:

"Although the effect of uncertainty in $H_2O$ data on $CO_2$ retrieval results could be also decreased by simultaneous retrieval of $H_2O$ with $CO_2$ in the TIR V1 algorithm, water vapor is abundant in the tropics, so that we cannot deny the possibility of its effect on $CO_2$ retrieval results. Similarly, error in the judgement of cloud contamination in low latitudes with high cloud occurrence frequency may affect $CO_2$ retrieval results."

*Page 10, lines 29-30, "The CME data that determined the bias-correction values of the $20°S-20°N$ latitude band were concentrated in East Asia": I was confused with this sentence. Please see my general comment #2.*

Reply:

As described above, we have replaced "East Asia" with "Southeast Asia" throughout the text. In the revised manuscript, we have listed specific airports (BKK, SIN, and CGK) where most CME data were obtained in the latitude band of 20°S−20°N as follows:

"The CME data that determined the bias-correction values of the 20°S−20°N latitude band were concentrated in Southeast Asia, as illustrated in Figure 1: BKK (Bangkok), SIN (Singapore), and CGK (Jakarta)."

We appreciate your comment.

*Page 10, line 34 – page 11, line 1, "in most areas at $0°−20°N$, and the negative biases were largest near airport locations in East Asia.": Same as above. Please see my general comment #2.*

Reply:

As described above, we have replaced "East Asia" with "Southeast Asia" throughout the text. We appreciate your comment.

*Page 11, lines 12-13, "More in-situ $CO_2$ data in the upper atmosphere in low latitudes": Hiaper Pole-to-Pole Observations (HIPPO) project observed latitudinal distributions of $CO_2$ concentrations in the free troposphere over the Pacific Ocean where mostly clean during 2009 to 2011 (e.g. Wofsy et al., 2011). The dataset has been used for transport model or satellite data validation (e.g. Wecht et al., 2012; Kulawik et al., 2013). The comparison with HIPPO data is out of the scope of this paper, but if the authors found some problems in using HIPPO data for validation, please write it in the discussion section or the introduction section.*

*Wofsy, S. C. et al.: HIAPER Pole-to-Pole Observations (HIPPO): fine-grained, global-scale measurements of climatically important atmospheric gases and aerosols, Phil. Trans. Roy. Soc. A: Math. Phys. Eng. Sci., 369, 2073–2086, doi:10.1098/rsta.2010.0313, 2011.*

Reply:

We agree with you. The reason why we did not use HIPPO data in this study is that HIPPO campaign observations were conducted for limited periods (October−November in 2009, March−April in 2010, June−July in 2011, and August−September in 2011, after starting the regular operation of GOSAT) in limited areas (mainly over the Pacific Ocean), so that they are not suitable for evaluating season- and latitude-dependent biases in GOSAT/TANSO-FTS TIR $CO_2$ data. As you pointed out, however, HIPPO data themselves are useful to validate $CO_2$ vertical profiles observed by satellite-borne sensors and simulated in models. Following your advice, we have touched on HIPPO data in the discussion part of the revised manuscript as follows:

"Although HIAPER Pole-to-Pole Observations (HIPPO) data (Wofsy et al., 2011) are not suitable for a comprehensive validation study as in this study due to their limited observation
periods, HIPPO $CO_2$ data are useful to validate $CO_2$ vertical profiles observed by
satellite-borne sensors and simulated in models (Kulawik et al., 2013)."
We appreciate your comment.
*Page 11, line 17, "Reconsideration of the setting of retrieval grid layers ...": Why do the authors think*
*the current setting of retrieval grid layers might not be suitable for retrievals and reconsideration might*
*solve it?*
Reply:
Total degree of freedom (defined as the trace of averaging kernel matrix) does not depend on the
setting of retrieval grid layers theoretically. In this situation, partial degree of freedom for each
retrieval grid layer (defined here as the diagonal element of averaging kernel matrix
corresponding to each retrieval grid layer, see Saitoh et al. (2016)) should decrease as the number
of retrieval grid layers increases. As illustrated in reference figure attached in Authors' reply to
Referee #2, the total degrees of freedom of GOSAT/TANSO-FTS TIR V1 $CO_2$ data are on
average 1.1−2.2 (depending on latitude and season), which means that we can derive information
on $CO_2$ concentrations in more than 1−2 vertical layers independently from observations by the
TIR band. In the TIR V1 Level 2 $CO_2$ retrieval algorithm, we have set 28 vertical grid layers.
Judging from the total degree of freedom of the TIR $CO_2$ data and the relatively small partial
degree of freedom for each vertical grid layer, we think we should reconsider the setting of
retrieval grid layers.
*Page 11, line 20, "during the JJA seasons of 2011 and 2011": Does this mean "2011 and 2012"?*
Reply:
We have modified the sentence. We appreciate you pointing out our mistake.
*Figs.3: The Y axis is described in altitude, not in pressure as seen in the following plots. For easy*
*reference, I would suggest adding a 2nd Y axis in pressure or adding a column in Table 1 to show*
*altitude [km] for each pressure levels. (Rough altitudes from International Standard Atmosphere or*
*the same kind might be enough for this purpose.*
Reply:
Following your suggestion, we have added a second vertical axis (y-axis) in pressure in Figure 3
of the revised manuscript. Here, we have taken pressure levels corresponding to the measurement
location of GOSAT/TANSO-FTS TIR data shown in the figure.

*Fig.4: Please replace "Altitude [km] in Y axis label with "Pressure [hPa]".*

Reply:

We have corrected the label of the vertical axis (y-axis) of Figure 4 of the revised manuscript. We appreciate you pointing out our mistake.

*Fig.7: I think drawing zero lines (i.e. no bias) in each panel makes the bias correction validity more visible.*

Reply:

Following your advice, we have drawn zero lines in each of the four panels of Figure 7 of the revised manuscript. We have also drawn zero lines in Figure 8 and 9 to show differences between each histogram clearly. We appreciate your suggestion.

*Fig.7 caption "Thick and dashed lines indicate the biases of the original TIR $CO_2$ data (no bias correction) and bias-corrected TIR $CO_2$ data, respectively.":*
*1. On my screen, all lines in each panel seem to have same line thickness. Do the authors mean "solid and dashed lines"?*
*2. This sentence does not match the main text which says that thick lines are bias-corrected values.*

Reply:

We appreciate you pointing out our mistake.
1. We have replaced "thick lines" with "solid lines" and exchanged "solid" for "dashed" in the caption for Figure 7 of the revised manuscript as follows:
"Dashed and solid lines indicate the biases of the original TIR $CO_2$ data (no bias correction) and bias-corrected TIR $CO_2$ data, respectively."
2. We have replaced "thick lines" with "solid lines" in the sentences related to Figure 7 in the revised manuscript.

*Fig.11, gray shade: Could the authors explain what gray zones in the figure are? (No data or out of color scale?)*

Reply:

Gray color in Figure 11 means no GOSAT/TANSO-FTS TIR $CO_2$ data in a 2.5° grid area. Following your advice, we have explained the meaning of gray color in the caption for Figure 11 of the revised manuscript as follows:
"There are no GOSAT/TANSO-FTS TIR $CO_2$ data in gray-shaded areas."

**Reply to Referee #2,**

*Overall, this is a good paper dealing with difficult but necessary bias corrections to TANSO-FTS*

*observations of mid-troposphere $CO_2$. It's a tricky subject, but the methodology is generally sound. However,*

*the paper is difficult to follow in some sections, and in many cases, the figures need some improvement and*

*clarification. I would recommend publication after some revisions in the text, and if the authors could better*

*address the issue of the number of layers in the forward model (see comment for page 10, line 32 below.)*

*General comment: Throughout the paper, the authors refer to the retrieval layers by number (layer 3, layer*

*4, etc.), rather than, say, its log mean pressure. These layer numbers are specific to their algorithm, and*

*referencing the layers by number is a little burdensome to the reader, even where the pressures are provided.*

*For example, Page 6, line 23 reads "Saitoh et al. (2016) showed that TIR V1 $CO_2$ data agreed well with*

*CME level flight $CO_2$ data in the UT region corresponding to retrieval layers 9 and 10." This would read*

*better if the pressures were given instead of the layer numbers. I suggest they prepare a table listing the*

*retrieval layer numbers, layer boundary pressures, and the log-mean pressures of the layers (similar to*

*Table 1 of Saitoh et al., 2016), and then just refer to a layer by its mean pressure rather than its number.*

Reply:

We greatly appreciate your comments. As described above, we have added Table 1 to show representative pressure levels of each of the retrieval grid layers used in the GOSAT/TANSO-FTS TIR V1 L2 $CO_2$ retrieval processing and referred to the retrieval grid layers by the representative pressure levels instead of retrieval grid numbers. In Table 1, we have kept the retrieval grid numbers for the convenience of TIR $CO_2$ data users. In the TIR V1 L2 $CO_2$ retrieval algorithm, we have calculated representative pressure level $P_{rlay}$, which is thermodynamically mean pressure level, by the following expression [Gallery et al., 1983]:

$$P_{rlay} = \{\frac{H_P H_\rho}{H_P + H_\rho}(P_{rlev\_j}\,\rho_{rlev\_j} - P_{rlev\_j+1}\,\rho_{rlev\_j+1})\}/\{H_\rho(\rho_{rlev\_j} - \rho_{rlev\_j+1})\}$$

$$H_P = \frac{-\Delta z}{\log_e(P_{rlev\_j+1}/P_{rlev\_j})}$$

$$H_\rho = \frac{-\Delta z}{\log_e(\rho_{rlev\_j+1}/\rho_{rlev\_j})}$$

$$\Delta z = \log_e \frac{P_{rlev\_j+1}}{P_{rlev\_j}} \times -\frac{Rd}{g} \times \frac{|T_{rlev\_j+1} - T_{rlev\_j}|}{2}$$

, where $P_{rlev\_j}$ and $P_{rlev\_j+1}$ are lower and upper pressure levels of each retrieval grid layer, respectively, $T_{rlev\_j}$ and $T_{rlev\_j+1}$ are temperatures at the two pressure levels, $\rho_{rlev\_j}$ and $\rho_{rlev\_j+1}$ are air densities at the two pressure levels, Rd is the gas constant, and g is the acceleration of gravity. Representative pressure levels change depending on temperature, which are stored in each of the

TIR V1 L2 $CO_2$ data files, but their variabilities are quite small. In Table 1, we have presented the averages of representative pressure levels of each retrieval grid layer calculated by using all GOSAT/TANSO-FTS measurements in 2010.

*Page 1, line 14: "…good spatial representability." It's not obvious what 'representability' means here. Would "resolution and precision" be a better phrase to use?*

Reply:

$CO_2$ concentrations in the free troposphere are well mixed compared to the concentrations near the surface and less affected by local point sources of $CO_2$; in that context, observations in the free troposphere can obtain $CO_2$ concentrations representative of regions, which can be dealt with in a global model estimating $CO_2$ surface fluxes. In the revised manuscript, we have modified the sentence to clarify this point as follows:

"$CO_2$ observations in the free troposphere can be useful for constraining $CO_2$ source and sink estimates at the surface due to their representativeness being away from local point sources of $CO_2$."

*Page 1, line 24: "(retrieval layers 5−6), …" It's not necessary to get into the details of their retrieval method in the abstract.*

Reply:

We have deleted the phrase in the abstract of the revised manuscript following your advice.

*Page 2, line 3: Suggest changing "(e.g., Gurney et al., 2002 Gurney et al., 2004)" to "(e.g., Gurney et al., 2002; 2004)".*

Reply:

Following your suggestion, we have modified the text in how to cite the references.

*Page 2, line 24: "spatial representability." Again, not obvious what it means here.*

Reply:

$XCO_2$ data obtained by measurements utilizing short-wave infrared (SWIR) band contained information on $CO_2$ concentrations near the surface compared to free tropospheric $CO_2$ measurements utilizing TIR band. However, satellite-borne sensors have relatively large field-of-views, and therefore their $XCO_2$ data are averaged concentrations in their field of views of several kilometers that are not too much affected by strong local point sources of $CO_2$. In the revised manuscript, we have modified the sentence as follows:

"Global $XCO_2$ data based on satellite observations are averaged concentrations in their field of views of several kilometers that are not too much affected by strong local point sources of $CO_2$, and have therefore been used to estimate surface $CO_2$ fluxes (Maksyutov et al., 2013; Saeki et al.,

2013a; Chevallier et al., 2014; Basu et al., 2013, 2014; Takagi et al., 2014)."

*Page 3, line 16: Suggest changing "...and has continued $CO_2$ and $CH_4$ operational measurements for*

*approximately eight years." to "and has continued operational measurements of $CO_2$ and $CH_4$ for*

*approximately eight years.*

Reply:

Following your suggestion, we have modified the sentence.

*Page 3, line 23: Suggest shortening "These studies showed the following: 1) TIR UT $CO_2$ data agreed..." to*

*"These studies showed: 1) TIR UT $CO_2$ data agreed..."*

Reply:

Following your suggestion, we have modified the sentence.

*Page 5, line 14: Suggest more explanation of why the averaging kernels are applied to the CME data and*

*then comparison made. This would be useful to the reader not well versed in averaging kernels etc.*

Reply:

Following your advice, we have added more explanation of why we should apply TIR $CO_2$

averaging kernel functions to CME aircraft profiles as follows:

"Observations by satellite-borne nadir-viewing sensors like TANSO-FTS have much lower vertical resolution than aircraft observations. Therefore, we smoothed the CME_obs. profile to fit its vertical resolution to the vertical resolution of corresponding TIR $CO_2$ profile by applying TIR

$CO_2$ averaging kernel functions (AK) to the CME_obs. profile, as follows (Rodgers and Connor,

2003):"

*Page 6, Section 4.2: It's not obvious why an "average" averaging kernel can be applied and not sometimes*

*be misleading. In addition to the effect of instrument parameters (SNR, spectral resolution, view angle etc.)*

*and assuming clear scenes only, the averaging kernel could vary by temperature gradient and thermal*

*contrast with the surface. How much does an averaging kernel vary within a grid box? It would help if the*

*authors briefly explain why they're using an averaged AK here and discuss the limitations of doing so.*

Reply:

We agree with you. TIR $CO_2$ averaging kernel functions depend on TIR measurement spectral noise, a priori $CO_2$ profile variability, and $CO_2$ Jacobians. In the TIR V1 L2 $CO_2$ retrieval algorithm, we set covariance matrices of the TIR measurement noise and a priori $CO_2$ profile in the same manner for all TIR $CO_2$ measurements, as described in Saitoh et al. (2016). The $CO_2$ Jacobians depend on temperature and $CO_2$ profiles, and therefore change with location and time. For a validation purpose based on one-by-one comparisons like TIR versus CME $CO_2$ profiles, we should apply corresponding TIR $CO_2$ averaging kernel functions, not averaged one. On the other hand, the purpose of comparisons between TIR and NICAM-TM $CO_2$ data is to evaluate the bias-correction values determined for each vertical layer, latitude band, and season. In addition, TIR $CO_2$ averaging kernel functions showed nearly identical structures with each other when collected for each $2.5°$ grid in one month, which means that applying the monthly averaged TIR $CO_2$ averaging kernel functions did not affect the conclusions of this study. From this standpoint, using monthly averaged TIR $CO_2$ averaging kernel functions instead of individual one is enough for our purpose. In the revised manuscript, we have added one paragraph in Section 4.2 and discussed the effect of using monthly averaged TIR $CO_2$ averaging kernel functions on our analysis. We appreciate your comments.

*Page 7, line 14 "In addition, negative biases of TIR $CO_2$ data against NICAM-TM $CO_2$ data increased by 1 ppm or less per year in all seasons, judging from the mode values, although the increase in negative biases was not evident in the comparisons over airports shown in Figure 6." I did not quite understand what is meant by this. Do they mean the bias varied by 1ppm or less?*

Reply:
We intended to say the following: negative biases of TIR $CO_2$ data against NICAM-TM $CO_2$ data seemed to increase over time, judging from each of the mode values for the three years and the rate of the increase was around and less than 1 ppm; however, the increase in the negative biases against NICAM-TM $CO_2$ data was not evident as was the case with the negative biases against CME $CO_2$ data discussed in Section 5.1. In the revised manuscript, we have modified the sentence as follows:
"In addition, negative biases of TIR $CO_2$ data against NICAM-TM $CO_2$ data in all seasons slightly increased over time, judging from the mode values, although the increase in negative biases was not much evident as in the comparisons over airports shown in Figure 6."

*Page 8, line 27: Typo: "… in the LT and ML regions." Did they mean "MT" regions?*

Reply:
We have modified the sentence. We appreciate you pointing out our mistake.

*Page 9, line 13: "As shown in Figure 6, the largest negative biases in TIR V1 $CO_2$ data existed in the MT*

*region in middle and low latitudes during spring and summer, where TANSO-FTS TIR measurements have*

*relatively large sensitivity to $CO_2$ concentrations and thus the retrievals are less constrained to a priori*

*concentrations." Some kind of comparison is in order to quantify the difference in $CO_2$ sensitivity here – say*

*average row-sum of averaging kernels, or total DOFS as a function of latitude.*

Reply:

We totally agree with you. We have modified the related sentences for consistency with the sentences in the second paragraph of Section 5.1, and then provided information on degrees of freedom of TIR V1 $CO_2$ data in low latitudes where the largest negative biases existed:

"As shown in Figure 6, the largest negative biases in TIR V1 $CO_2$ data existed in the MT region in low latitudes (20°S−20°N) during the JJA season. Degrees of freedom (DF) of TIR V1 $CO_2$

data were highest in low latitudes, exceeding 2.2 in all seasons, which means retrieved $CO_2$

concentrations there contained more information coming from TANSO-FTS TIR L1B spectra and thus were relatively less constrained to a priori concentrations."

The DF values have been referred from the below figure that shows monthly averaged DF values for each 10° latitude in January (blue), April (green), July (red), and October (light blue) in 2010.

[Figure]

Reference figure. Monthly averaged DF values of TIR V1 $CO_2$ data for each 10° latitude in

January, April, July, and October 2010, shown by blue, green, red, and light blue lines, respectively. Here, GOSAT/TANSO-FTS observations with high elevated areas (surface pressure less than 736 hPa) were excluded.

*Page 9, line 15: "This implies that biases in L1B spectra are a major cause of the negative biases in*

*retrieved $CO_2$ concentrations, as Saitoh et al. (2016) noted in the UT region." The wording is confusing.*

*Does this mean there are biases in the L1b radiances related to latitude and season, or are there fitting*

*biases from the retrieval algorithm? Judging from the rest of the paragraph where the authors write about*

*retrieval of surface parameters, I think they're referring to fitting bias, but whatever the bias is, it should be explicitly described.*

Reply:

According to comparisons between TANSO-FTS TIR and S-HIS radiance spectra (Kataoka et al., 2014) and theoretical radiance error estimations (Kuze et al., 2016), TANSO-FTS TIR L1B radiance spectra had considerable biases. In low latitudes, retrieved $CO_2$ data contained more information coming from TANSO-FTS TIR L1B spectra judging from their highest DF values. This means that the effect of the L1B radiance biases should be also largest in TIR $CO_2$ data in low latitudes. The magnitude of the TIR L1B radiance biases may change by scene, but we have not yet drawn any conclusion on the dependence of the radiance biases on time, location, viewing angle, thermal condition of TANSO-FTS instrument, and so on. As the related three paragraphs in Discussion were less organized, we have reorganized the discussion on the relation between L1B radiance biases and L2 $CO_2$ negative biases against CME $CO_2$ data in the revised manuscript.

*Page 10, line 4: "From these results, we conclude that using the 10-μm band in conjunction with the 15-μm and 9-μm bands in the V1 retrieval algorithm is a probable cause of the negative biases in retrieved $CO_2$ concentrations in the LT and MT regions." While I don't disagree with this, this would be more convincing if the authors compared their results using the different mixes of $CO_2$ bands directly against the aircraft measurements.*

Reply:

We totally agree with you. We have also showed nearby CME $CO_2$ profiles by gray lines in Figure 10 of the revised manuscript other than TIR $CO_2$ retrieval results. We appreciate your suggestion.

*Page 10, Line 13: "According to Figure 13 in Kuze et al. (2016), there was no distinct uncertainty in the 10-μm band in the latest version of the TANSO-FTS TIR spectra." The wording of this leaves me uncertain of what they're claiming. Uncertainty of linestrengths or low fitting residual? Are they saying that using the 10 micron band of $CO_2$ does not add significant bias? This should be clarified.*

Reply:

Kuze et al. (2016) performed theoretical estimation of radiance biases of TANSO-FTS TIR L1B V161 and newer version V201 spectra. The radiance biases inherent in the TANSO-FTS TIR L1B spectra were attributable to several calibration issues, mainly due to polarization correction. According to theoretical calculations shown in Figure 13 in Kuze et al. (2016), there were no distinct radiance biases in the 10-μm band (930−990 cm$^{-1}$) in the latest version of the

TANSO-FTS TIR spectra. If it is true for observed TIR radiances, our test retrievals imply that simultaneous retrieval of surface parameters for TIR spectra at the 10-μm band with less radiance bias worsened $CO_2$ retrieval results. We have clearly stated this in the revised manuscript.

*Page 10, paragraph beginning line 17: As noted earlier, it would really help the reader if the authors*

*referred to the retrieval layers by pressure and not layer number.*

Reply:

Following your advice, we have referred to the lower and upper pressure levels of the two retrieval grid layers that we focused on.

*Page 10, line 32: "In retrieval from TIR spectra, the more atmospheric layers in which we retrieve $CO_2$*

*concentrations, the lower the information content of the retrieval result in each layer becomes; as a result,*

*the retrieved concentrations are constrained by a priori model data. Thus, there is a high possibility of large*

*biases in retrieved TIR $CO_2$ concentrations in low latitudes." This assertion needs to be tested. It is true that*

*with more layers, the information is spread out more, but the overall information content, as measured by*

*the degrees-of-freedom-of-signal (trace of the averaging kernel) can be the same or very similar, as can the*

*retrieved profiles (depending on what the off-diagonals are for the a priori background covariance.) It's*

*quite possible that if the background a priori is biased, then a TIR retrieval can also be biased not because*

*of the number of retrieval layers, but, particularly at low latitudes, because of water vapor interference,*

*undetected boundary- layer clouds changing the thermal contrast with the surface, or biases in the*

*temperature. Again, this needs to be tested, or the statement removed or at least reworded as a*

*hypothesizing.*

Reply:

We totally agree with you. Our wording in the original manuscript leads to misunderstanding. We here intended to say that TIR $CO_2$ retrieval were somewhat constrained by a priori concentrations.

In the MT region in low latitudes, a priori $CO_2$ concentrations taken from the NIES-TM05 model probably have larger uncertainties due to the parameterization of vertical transport. Therefore, there is a possibility of more biases attributed to the a priori uncertainties in retrieved TIR $CO_2$

data there. Following your suggestion, we have removed the related statement and modified the sentences in the revised manuscript as follows:

"In low latitudes, there are relatively strong updrafts, and thus there are larger uncertainties among models than in other areas due to differences in the parameterization of vertical transport.

Therefore, a priori $CO_2$ concentrations taken from the NIES-TM05 model (Saeki et al., 2013b)

probably have larger uncertainties in the MT region in low latitudes. As retrieved TIR $CO_2$

concentrations were to some extent constrained by a priori concentrations, they possibly had more biases attributed to the a priori uncertainties in the MT region in low latitudes."

We greatly appreciate your comment.

*Figure 5: It would be much clearer to the reader if they provided guidance to the different panels and lines*

*in a legend box on the figure, rather than only in the caption. It would also help, for a reader skimming the*

*paper, to describe what "CME_AK $CO_2$" means in the caption as well as the text of the paper.*

Reply:

Following your advice, we have provided information on seasons in each panel and described each line in both left and right sides of the panel (a). In the caption of the revised manuscript, we have described what CME_AK $CO_2$ means as follows:

"The CME_AK $CO_2$ data are CME $CO_2$ data to which TIR $CO_2$ averaging kernel functions are applied."

*Figure 6: Use pressures and not layer numbers on vertical axis. It would also be better if latitude*

*information and season (line color) were provided as a legend on the figure. It would help if the lines in the*

*top panels had slight vertical offsets to clarify how different the error bars are from each other.*

Reply:

Following your advice, we have presented the representative pressure levels of the six retrieval grid layers shown in Table 1 instead of their layer numbers. We have provided information on latitude bands and colors for seasons as a legend and slightly shifted horizontal bars for 1-σ

standard deviations in Figure 6 of the revised manuscript. We appreciate your comments.

*Figure 7: It's not clear here (or in the text) at what pressures they are comparing avg $CO_2$ with NICAM. The*

*contrast between the mid-gray and light-gray lines is not enough to easily distinguish between them.*

Reply:

Figure 7 includes all comparison results between TIR and NICAM-TM $CO_2$ data in the six retrieval grid layers from 736 to 287 hPa (retrieval layers 3−8). In the revised manuscript, we have stated this clearly in the revised manuscript as follows:

"Figure 7 shows the frequency distributions of differences in monthly averaged $CO_2$

concentrations between TIR and NICAM-TM $CO_2$ data in all retrieval layers from 736 to 287

hPa in all 2.5° grids over the latitude range of 40°S to 60°N.".

Following your advice, we have presented the lower and upper pressure levels of the six retrieval layers that we focused on and used red and blue colors instead of light-gray and mid-gray colors in Figure 7 of the revised manuscript. We appreciate your comments.

*Figure 8: Please use pressures instead of layer numbers. Again, the contrast between the mid-gray and*

*light-gray lines is not enough to easily distinguish between them.*

Reply:

Following your advice, we have presented the lower and upper pressure levels of each set of the six retrieval grid layers that we focused on and used red and blue colors instead of light-gray and mid-gray colors in Figure 8 of the revised manuscript.

*Figure 9: Again, please state the pressures instead of "layer 7-8."*

Reply:

Following your advice, we have modified Figure 9 to present the lower and upper pressure levels of the two retrieval grid layers that we focused on.

*Figure 10: Please also describe the lines and the location/times the different panels represent as a legend*

*rather than just in the caption.*

Reply:

Following your advice, we have modified Figure 10: we have separated the two results of Figure

10(b) and discarded the result of Figure 10(a) of the original manuscript to simplify the figure, provided information on the locations (both over Narita airport) and dates ((a) April 1, 2010 and (b) April 30, 2010) of the two results in the caption and each of the panels, and described each of the five lines in the panel (b).

In the revised manuscript showing the changes made that is attached below, we have showed the changes that we made to address the comments by Referee #1 by blue color, and the changes that we had made to address the comments by Referee #2 and have been already reflected in AMTD by red color.

[revised manuscript text omitted]
 1, 2010A case in low latitudes in summer (July) and and (b) atwo cases of April 30, 2010in northern middle latitudes in spring (April).

[Figure]

Figure 11. Latitude–longitude cross-sections of differences in monthly averages of GOSAT/TANSO-FTS TIR CO$_2$ data and NICAM-TM CO$_2$ data with considering TIR CO$_2$ averaging kernel functions (TIR minus NICAM-TM) in July 2010. The upper and lower panels show the results on 682 hPa (in retrieval layer 3 (736–631 hPa) and 314 hPa8 (retrieval layer 8341–287 hPa), respectively. There are no GOSAT/TANSO-FTS TIR CO$_2$ data in gray-shaded areas.

---

## Author Response (AR2)

**Reply to the comments by Editor**
* * *
**To Associate Editor**

We appreciate you reading our revised manuscript carefully and giving valuable suggestions for revisions. We have modified the text following your suggestions.

*1) Abstract, 1$^{st}$ sentence. This is still a bit confusing - maybe say:*
*"$CO_2$ observations in the free troposphere can be useful for constraining $CO_2$ source and sink estimates at the surface since they represent $CO_2$ concentrations away from point source emissions."*

    Reply:
    We have modified the sentence following your suggestions. We appreciate your suggestion.

*2) p. 2 line 23: Maybe say:*
*"Global $XCO_2$ data, based on satellite observations, are averaged concentrations over fields of view that typically cover several kilometers. This spatial resolution is not sufficient for measuring individual strong local point sources of $CO_2$."*

    Reply:
    Following your suggestion, we have modified the text as follows:

[revised manuscript text omitted]
|---|---|---|---|---|---|---|---|---|---|
| 2010 | | -2.0 | 0.5 | -2.5 | 0.0 | -2.5 | 0.0 | -2.5 | 0.5 |
| | | 13.6 | 13.9 | 10.5 | 12.9 | 10.7 | 10.4 | 11.8 | 11.1 |
| | | 641,427 | | 947,983 | | 1,176,998 | | 1,279,370 | |
| 2011 | | -3.0 | 0.5 | -3.5 | 1.0 | -2.5 | 1.0 | -2.5 | 0.5 |
| | | 11.3 | 12.1 | 8.8 | 11.4 | 9.8 | 9.4 | 11.5 | 9.4 |
| | | 1,156,444 | | 1,093,808 | | 1,156,010 | | 1,222,288 | |
| 2012 | | -3.0 | 0.0 | -4.0 | 0.0 | -3.5 | 1.0 | -4.0 | 0.5 |
| | | 12.1 | 13.1 | 8.7 | 11.8 | 9.3 | 10.5 | 10.6 | 10.5 |
| | | 1,050,530 | | 1,010,457 | | 1,148,979 | | 1,117,909 | |

[revised manuscript text omitted]